# A Bayesian Approach to Segmentation with Noisy Labels via Spatially Correlated Distributions

**Ryu Tadokoro**                                                                    *tadokororyuryu@gmail.com*
*Tohoku University*

**Tsukasa Takagi**                                                                  *takagi@preferred.jp*
*Preferred Networks, Inc.*

**Shin-ichi Maeda**                                                                 *ichi@preferred.jp*
*Preferred Networks, Inc.*

**Reviewed on OpenReview:** *https://openreview.net/forum?id=oMgfr8Kk2x*

## Abstract

In semantic segmentation, the accuracy of models heavily depends on the high-quality annotations. However, in many practical scenarios, such as medical imaging and remote sensing, obtaining true annotations is not straightforward and usually requires significant human labor. Relying on human labor often introduces annotation errors, including mislabeling, omissions, and inconsistency between annotators. In the case of remote sensing, differences in procurement time can lead to misaligned ground-truth annotations. These label errors are not independently distributed, and instead usually appear in spatially connected regions where adjacent pixels are more likely to share the same errors. To address these issues, we propose an approximate Bayesian estimation based on a probabilistic model that assumes training data include label errors, incorporating the tendency for these errors to occur with spatial correlations between adjacent pixels. However, Bayesian inference for such spatially correlated discrete variables is notoriously intractable. To overcome this fundamental challenge, we introduce a novel class of probabilistic models, which we term the **ELBO-Computable Correlated Discrete Distribution (ECCD)**. By representing the discrete dependencies through a continuous latent Gaussian field with a Kac-Murdock-Szegö (KMS) structured covariance, our framework enables scalable and efficient variational inference for problems previously considered computationally prohibitive. Through experiments on multiple segmentation tasks, we confirm that leveraging the spatial correlation of label errors improves robustness. Notably, in specific tasks such as lung segmentation, the proposed method achieves performance comparable to training with clean labels under moderate noise levels. Code is included in the supplementary materials.

## 1 Introduction

Semantic segmentation, which involves classifying each pixel in an image into one of several classes, is a crucial task in computer vision. In supervised learning, the accuracy of segmentation models critically depends on the quality of the annotations in the training data. However, obtaining truly accurate pixel-level annotations is challenging in many practical applications. Even when expert annotators are employed, errors, omissions, and subjectivity in interpretation are inevitable, leading to inconsistencies in the datasets. In particular, high inter- and intra-annotator variability is widely reported in medical imaging, where experts may have differing interpretations of the same structures. For instance, several studies Zhang et al. (2020a); Lampert et al. (2016); Rädsch et al. (2023); Yang et al. (2023) highlight significant discrepancies among expert annotators; some delineate structures more generously, while others prefer more conservative annotations. Observer-dependent annotations exacerbate label noise in supervised learning. Label noise is also a critical

issue in remote sensing, where determining ground truth labels often requires field surveys over large and sometimes inaccessible regions Frénay & Verleysen (2013); Pelletier et al. (2017); Foody (2002). Due to the logistical and economic challenges of large-scale ground truth collection, researchers frequently rely on automatic labeling systems, which may introduce systematic errors. Additionally, high-quality annotated datasets remain a critical bottleneck for supervised learning, particularly in remote sensing applications where annotations are often repurposed across different types of satellite images. For example, the OpenEarthMap dataset Xia et al. (2023) was created by manually annotating high-resolution optical satellite images for semantic segmentation. However, these annotations are sometimes reused for synthetic aperture radar (SAR) imagery, despite differences in resolution and capture conditions Huang et al. (2021); Zhang et al. (2021); Liu et al. (2023). Additionally, changes in artificial structures or variations in land cover further contribute to label inconsistencies Fritz et al. (2009); Huang et al. (2021).

Various approaches have been proposed to mitigate the adverse effects of noisy labels. Some methods attempt to stop training early to prevent the network from overfitting to noise and generating unreliable pseudo-labels Liu et al. (2024; 2022), while others modify the loss function to be more robust against large errors Gonzalez-Jimenez et al. (2025). Although these techniques can reduce the influence of noisy annotations, they do not fundamentally address the core reason why the standard supervised learning framework fails in the presence of noisy labels.

We propose a method that directly tackles the root cause of this issue. To understand this root cause, we first revisit the probabilistic assumptions underlying the standard cross-entropy loss. In supervised learning for segmentation models, training typically reduces to optimizing this loss function. This optimization implicitly follows a maximum likelihood estimation (MLE) framework under the assumption that the training data consists of independent and identically distributed samples drawn from the joint distribution of images and clean labels. However, when labels are noisy, the assumption of identically distributed data no longer holds, as the observed labels systematically deviate from clean labels, leading to performance degradation in the trained model.

To address this issue, we maintain the MLE framework but generalize the underlying probabilistic model to make it more suitable for real-world scenarios with noisy labels. Specifically, we introduce a model that explicitly accounts for the presence of noisy labels, which differ from the clean labels due to labeling errors. In practice, annotation errors tend to exhibit strong spatial correlations - mislabeling often occurs in contiguous regions. Variations in annotation criteria among experts, as well as changes in the underlying scene — such as the construction or demolition of buildings or alterations in vegetation — further reinforce this spatial dependency. Given this, we assume that label errors are not independent but instead exhibit spatial correlations among pixels.

However, directly modeling spatial correlations among discrete labels leads to an intractable marginal likelihood, a long-standing challenge in probabilistic modeling Koller & Friedman (2009); Sutton & McCallum (2012). Our core contribution is to resolve this challenge by proposing a novel class of tractable models, the **ELBO-Computable Correlated Discrete Distribution (ECCD)**. This discrete distribution is represented through continuous variables that follow a Gaussian distribution, where spatial correlations between pixels are expressed via a covariance matrix. Representing discrete variables through a Gaussian distribution successfully circumvents the intractability of summing over all possible realizations of the discrete variables. While the covariance matrix, whose number of elements scales quadratically with the number of pixels, introduces additional computational challenges in evaluating the ELBO, particularly in computing second-order statistics, its inverse, and its determinant. To overcome these computational intractabilities, we leverage the Kac-Murdock-Szegö (KMS) matrix Fikioris (2018); Kac et al. (1953), which enables efficient computations necessary for ELBO evaluation. For a detailed discussion on the theoretical properties of our model, including its relation to Logit-Gaussian Random Fields and classical MRF/CRF literature, please refer to Appendix B. To validate the effectiveness of our approach, we conduct extensive empirical evaluations on multiple segmentation tasks. Our experimental results demonstrate that our method significantly improves robustness against label noise, particularly in scenarios with moderate to high levels of spatially correlated label noise. In summary, our contributions are as follows:

1. **A Novel Class of Tractable Distributions for Correlated Discrete Variables:**
   We introduce the **ECCD**, a new class of probabilistic models for tractable inference on high-dimensional discrete variables with stationary, exponentially decaying correlations—a structure common to spatial grids and temporal sequences. In this work, we apply the ECCD to learn from spatially correlated label noise in medical and remote sensing, where the model circumvents computational intractability by representing discrete dependencies via a continuous latent Gaussian field.

2. **Efficient ELBO optimization via the KMS matrix:**
   The ECCD's covariance matrix introduces computational challenges that scale quadratically with the number of pixels. We overcome this by leveraging the Kac-Murdock-Szegö (KMS) matrix. To the best of our knowledge, this is the first work to introduce the KMS matrix as a scalable computational tool for variational inference with correlated latent variables, enabling efficient computation of the key operations necessary for optimizing the ELBO of the ECCD.

3. **Extensive empirical validation:**
   We perform comprehensive experiments on multiple segmentation tasks, demonstrating that our method significantly improves robustness to label noise. Our approach outperforms existing techniques, particularly in scenarios with moderate to high levels of spatially correlated label noise, validating the effectiveness of our probabilistic formulation.

## 2 Related Work

Learning with noisy labels has been a significant focus in the field of machine learning, particularly classification tasks Patrini et al. (2017); Yao et al. (2020); Han et al. (2018); Wei et al. (2020). One research direction models the relationship between the clean label $y^*$ and the noisy label $y$ through a transition matrix Patrini et al. (2017); Yao et al. (2020) that characterizes the probabilities from the clean label to the noisy label. For example, the transition probability from a clean label of $k$ to a noisy label of $j$ is represented as $p(y = j|y^* = k)$. Additionally, methods have been proposed to estimate an instance-dependent transition matrix Xia et al. (2020); Berthon et al. (2021), where the transition probabilities depend on the instance $x$ also, denoted as $p(y = j|y^* = k, x)$. These classification methods can be adapted for semantic segmentation tasks; however, they treat label errors as being independent to each pixel, thereby ignoring the spatial correlations in label errors that may be present in annotations. This limitation is particularly critical in applications such as medical imaging Zhang et al. (2020a); Shi & Wu (2021); Zhang et al. (2020b) and remote sensing Mnih & Hinton (2012); Song et al. (2022); Liu et al. (2024); Henry et al. (2021), where different types of noise may occur.

Other lines of research include work by Zhang *et al.* Zhang et al. (2020c), which proposed a Tri-Network framework that trains three networks simultaneously, with each pair selecting reliable pixels based on their loss maps, thereby achieving robust learning even with coarse annotations. Liu *et al.* Liu et al. (2022) introduced a mechanism to detect the moment when a network begins to overfit noisy labels. Gonzalez *et al.* Gonzalez-Jimenez et al. (2025) designed T-Loss based on a Student-t distribution to apply a logarithmic penalty on large errors, reducing undue influence from outlier pixels. Nonetheless, these methods do not take into account the spatial correlations in label errors that may occur within annotations.

Unlike the aforementioned related works, there are few studies that consider spatial correlations. Li *et al.* Li et al. (2021) leveraged superpixel segmentation to group pixels and smooth the network outputs within each superpixel, thereby indirectly incorporating spatial context. This method, however, relies heavily on the quality of the generated superpixels. More recently, Yao *et al.* Yao et al. (2023) proposed an approach that explicitly models spatial correlations by applying a Markov process that consists of expansion and shrinkage steps of the annotation masks. Although this method achieves better performance, it is mainly limited to addressing noise along the boundaries. In real-world scenarios, segmentation labels are often corrupted by a diverse range of noise types, including omission noise, misalignment noise, boundary uncertainties, and other systematic labeling errors Jiménez & Decaestecker (2024); Vădineanu et al. (2022); Liu et al. (2024); Mnih & Hinton (2012); Jiang et al. (2021).

Our proposed probabilistic model can incorporate spatial correlations in label errors without making assumptions about the noise types, thereby handling a wide range of noise types.

## 3 Problem Formulation

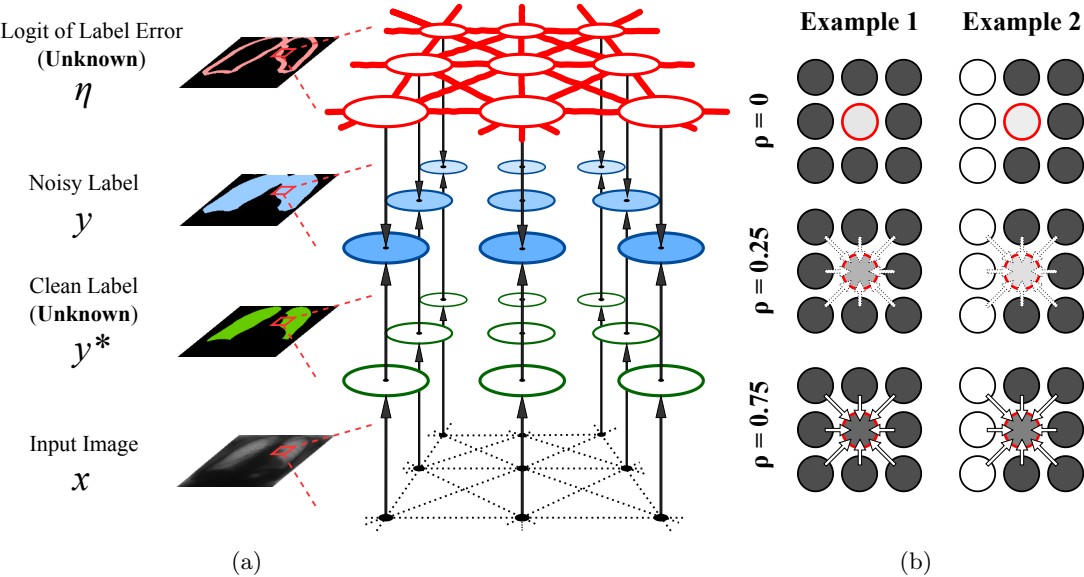

(a)           (b)

Figure 1: **(a) Graphical model of our proposed method, the ECCD.** The model introduces a latent Gaussian variable $\boldsymbol{\eta}$ that encodes spatial correlations of label error among adjacent pixels, thereby enabling a more realistic modeling of the dependency between pixels.
**(b) Conceptual illustration of spatial correlation.** Examples 1 and 2 show how varying $\rho$ affects the correlation strength between the center pixel and its neighbors (top, bottom, sides, and diagonals). When adjacent pixels are likely to contain label errors and spatial correlation is strong ($\rho$ is high) label errors can be inferred with high confidence, even if they are rare.

### 3.1 Probabilistic Foundations of Standard Supervised Segmentation

In standard supervised learning for semantic segmentation, the training process is typically framed as the minimization of the cross-entropy loss. This objective function is not arbitrary; it is derived from the principle of Maximum Likelihood Estimation (MLE) under a specific, and often implicit, probabilistic model. This underlying probabilistic model makes two fundamental assumptions: (1) the provided annotations $\mathbf{y}$ are the true, noise-free clean labels $\mathbf{y}^*$, and (2) the label of each pixel $y_i$ is conditionally independent of the labels of other pixels given the input image $\mathbf{x}$.

Under these assumptions, the goal is to find the model parameters $\theta$ that maximizes the log-likelihood of the observed data:

$$\log p_\theta(\mathbf{y}|\mathbf{x}) = \log \prod_{i=1}^{HW} p_\theta(y_i|\mathbf{x}) = \sum_{i=1}^{HW} \log p_\theta(y_i|\mathbf{x}), \tag{1}$$

where $HW$ is the total number of pixels. The segmentation model $p_\theta(y_i|\mathbf{x})$ outputs a probability distribution over the set of possible classes $\mathcal{Y}$ for each pixel $i$. Let this distribution be represented by a vector of probabilities $\{p_\theta(y_{ik}|\mathbf{x})\}_{k=1}^{|\mathcal{Y}|}$. If we represent the ground-truth label $y_i$ as a one-hot vector, where $y_{ik} = 1$ if pixel $i$ belongs to class $k$ and $y_{ik} = 0$ otherwise, the log-likelihood for a single pixel can be written as

$\sum_{k=1}^{|\mathcal{Y}|} y_{ik} \log p_\theta(y_{ik}|\mathbf{x})$. The total log-likelihood for the image is therefore:

$$\mathcal{L}_{\text{MLE}}(\theta) = \sum_{i=1}^{HW} \sum_{k=1}^{|\mathcal{Y}|} y_{ik} \log p_\theta(y_{ik}|\mathbf{x}). \tag{2}$$

Minimizing the negative of this quantity, $-\mathcal{L}_{\text{MLE}}(\theta)$, is precisely equivalent to minimizing the standard multi-class cross-entropy loss. This derivation highlights that the validity of the cross-entropy loss is contingent on the assumption that the training labels are clean and pixel-wise independent samples from the true data distribution.

## 3.2 A General Probabilistic Model for Learning with Noisy Labels

In many practical scenarios, particularly in medical imaging and remote sensing, the assumption that observed labels $\mathbf{y}$ are identical to the true clean labels $\mathbf{y}^*$ does not hold. Annotations are often corrupted by noise, which can be spatially correlated. To address this, we generalize the probabilistic model by explicitly distinguishing between the observed labels $\mathbf{y}$ and the unobserved (latent) clean labels $\mathbf{y}^*$. This leads to the objective of maximizing the marginal log-likelihood, which integrates out the uncertainty over the unknown clean labels $\mathbf{y}^*$:

$$\log p_\theta(\mathbf{y}|\mathbf{x}) = \log\left(\sum_{\mathbf{y}^*} p_\theta(\mathbf{y}, \mathbf{y}^*|\mathbf{x})\right) = \log\left(\sum_{\mathbf{y}^*} p(\mathbf{y}|\mathbf{y}^*)p_\theta(\mathbf{y}^*|\mathbf{x})\right). \tag{3}$$

Here, $p_\theta(\mathbf{y}^*|\mathbf{x})$ is the probability of the clean labels $\mathbf{y}^*$ given the image $\mathbf{x}$, which is modeled by the segmentation network, and $p(\mathbf{y}|\mathbf{y}^*)$ represents the label noise process.

This formulation is a direct generalization of the standard MLE objective described in Section 3.1. To see this, consider the noise-free case where the noise model $p(\mathbf{y}|\mathbf{y}^*)$ collapses to a Kronecker delta function $\delta(\mathbf{y}^*, \mathbf{y})$, which equals 1 if and only if $\mathbf{y}^* = \mathbf{y}$, and is zero otherwise. In this scenario, the summation in Eq. (3) contains only one non-zero term, and the objective reduces precisely to maximizing $\log p_\theta(\mathbf{y}|\mathbf{x})$, which is the basis for the standard cross-entropy loss as shown in Eq. (1). However, the summation over all possible configurations of $\mathbf{y}^*$ is computationally intractable, as the number of terms ($|\mathcal{Y}|^{HW}$) grows exponentially with the number of pixels. One way to overcome this is to assume a pixel-independent noise process, $p(\mathbf{y}|\mathbf{y}^*) = \prod_i p(y_i|y_i^*)$, but this fails to capture real-world spatially correlated label noise, as mentioned in Section 1.

## 4 The Proposed ECCD Framework

### 4.1 A Tractable Variational Objective via the ECCD

To overcome the intractability of the marginal likelihood while modeling spatial correlations, we resort to variational inference and maximize its Evidence Lower Bound (ELBO). We introduce a continuous latent variable, $\boldsymbol{\eta} \in \mathbb{R}^{HW}$, which represents the logit of the label error probability for each pixel. This variable captures spatial correlations and indirectly imposes spatial dependencies on the labels as follows:

$$p(\mathbf{y}|\mathbf{y}^*) = \int p(\mathbf{y}|\mathbf{y}^*, \boldsymbol{\eta})p(\boldsymbol{\eta})d\boldsymbol{\eta} = \int \left\{\prod_i p(y_i|y_i^*, \eta_i)\right\} p(\boldsymbol{\eta})d\boldsymbol{\eta}. \tag{4}$$

Figure 1a depicts the graphical model illustrating this assumption. To fully specify this generative model, which is the central component of the ECCD, we define its constituent distributions. The prior $p(\boldsymbol{\eta})$ is modeled as a Gaussian, $p(\boldsymbol{\eta}) = \mathcal{N}(\boldsymbol{\eta}|\boldsymbol{\mu}, \boldsymbol{\Sigma})$, with its covariance matrix structured using a Kac-Murdock-Szegö (KMS) matrix Fikioris (2018); Kac et al. (1953), which we detail in Section 4.2. The proposed model can be explicitly characterized as a Logit-Gaussian Random Field. While general inference in such models is intractable, our specific restriction to the KMS structure enables efficient variational inference, which we detail in Appendix B.1. The conditional label noise model $p(y_i|y_i^*, \eta_i)$ is defined as:

$$p(y_{ik} = 1|y_{ic}^* = 1, \eta_i) = \delta_{kc}(1 - r(\eta_i)) + (1 - \delta_{kc})r(\eta_i)B_{kc}, \tag{5}$$

where $\delta_{kc}$ is the Kronecker delta, $r(\cdot)$ is the sigmoid function that maps the latent variable $\eta_i$ to a label error probability, and $B$ is a learnable class transition matrix that specifies the probability distribution over incorrect labels when an error occurs, with its elements satisfying $B \in [0,1]^{|\mathcal{Y}| \times |\mathcal{Y}|}$, $B_{cc} = 0$ and $\sum_{k=1}^{|\mathcal{Y}|} B_{kc} = 1$, for each true class $c \in \mathcal{Y}$.

To derive a tractable objective for this model, we introduce a variational distribution $q(\mathbf{y}^*, \boldsymbol{\eta} | \mathbf{y})$ to approximate the true posterior. We assume a factorized form symmetric to the generative model: $q(\mathbf{y}^*, \boldsymbol{\eta} | \mathbf{y}) = q(\boldsymbol{\eta}) \prod_i q(y_i^* | y_i, \eta_i)$. The variational posterior for $\boldsymbol{\eta}$ is also a Gaussian, $q(\boldsymbol{\eta}) = \mathcal{N}(\boldsymbol{\eta} | \boldsymbol{m}, \boldsymbol{\Gamma})$, and the conditional posterior for the clean labels is parameterized as:

$$q(y_{ik}^* = 1 | y_{ic} = 1, \eta_i) = \delta_{kc}(1 - r(\eta_i)) + (1 - \delta_{kc})r(\eta_i)A_{kc}, \tag{6}$$

where $A \in [0,1]^{|\mathcal{Y}| \times |\mathcal{Y}|}$ is another learnable class transition matrix modeling the inverse noise process (i.e., inferring the clean label from the noisy one), with analogous constraints ($A_{cc} = 0$ and $\sum_{k=1}^{|\mathcal{Y}|} A_{kc} = 1$ for each noisy class $c \in \mathcal{Y}$). With the generative model and variational posterior fully defined, we can derive the Evidence Lower Bound (ELBO) as follows (see Appendix A.2 for details):

$$\begin{aligned}
\log p(\mathbf{y}|\mathbf{x}, \boldsymbol{\theta}) \geq{} & \mathbb{E}_{q(\mathbf{y}^*, \boldsymbol{\eta}|\mathbf{y})} \left[ \log p(\mathbf{y}|\mathbf{y}^*, \boldsymbol{\eta}, \mathbf{x}, \boldsymbol{\theta}) \right] - KL \left[ q(\mathbf{y}^*, \boldsymbol{\eta}|\mathbf{y}) \, \| \, p(\boldsymbol{\eta}, \mathbf{y}^*|\mathbf{x}, \boldsymbol{\theta}) \right] \\
={} & \sum_i \mathbb{E}_{q(\eta_i)q(y_i^*|y_i,\eta_i)} \left[ \log p_{\boldsymbol{\theta}}(y_i^*|\mathbf{x}) \right] - \mathbb{E}_{q(\boldsymbol{\eta})} \left[ \log q(\boldsymbol{\eta}) \right] + \mathbb{E}_{q(\boldsymbol{\eta})} \left[ \log p(\boldsymbol{\eta}) \right] \\
& - \sum_i \mathbb{E}_{q(\eta_i)q(y_i^*|y_i,\eta_i)} \left[ \log q(y_i^*|y_i, \eta_i) \right] + \sum_i \mathbb{E}_{q(\eta_i)q(y_i^*|y_i,\eta_i)} \left[ \log p(y_i|y_i^*, \eta_i) \right].
\end{aligned} \tag{7}$$

A key advantage of this formulation is that the expectations over the discrete latent labels $\mathbf{y}^*$ in Eq. (7) are now decomposed into a sum over individual pixels. This avoids the intractable summation over all possible label map configurations that hindered the direct optimization of the marginal likelihood in Eq.(3), rendering the objective tractable. The overall goal is to optimize the model parameters $\boldsymbol{\theta}$ and the variational parameters of $q(\boldsymbol{\eta})$ and $q(y_i^* | y_i, \eta_i)$ to maximize this lower bound. The ELBO in Eq. (7) has an important interpretation as a principled generalization of the standard cross-entropy loss. To see this, we can analyze its components.

The first term in this ELBO in Eq. (7) can be interpreted as an expected reconstruction term. By expanding the expectation over the discrete clean label $y_i^*$, it becomes a cross-entropy loss with soft labels:

$$\sum_i \mathbb{E}_{q(y_i^*|y_i,\eta_i)q(\eta_i)} \left[ \log p_{\boldsymbol{\theta}}(y_i^*|\mathbf{x}) \right] = \sum_{i=1}^{HW} \sum_{k=1}^{|\mathcal{Y}|} q(y_{ik}^* = 1|y_i) \log p_{\boldsymbol{\theta}}(y_{ik}^* = 1|\mathbf{x}), \tag{8}$$

where $q(y_{ik}^* = 1|y_i) = \mathbb{E}_{q(\eta_i)}[q(y_{ik}^* = 1|y_i, \eta_i)]$ is the marginal posterior probability that the clean label for pixel $i$ is class $k$. This replaces the hard, fixed one-hot observed labels with soft, inferred posterior probabilities. Building on this interpretation of the first term, we can now demonstrate more formally that our entire objective is a principled generalization of the standard cross-entropy loss. This is shown by examining the conditions under which the remaining terms in the ELBO vanish.

The second and third terms together constitute the negative Kullback-Leibler (KL) divergence, $-\mathrm{KL}[q(\boldsymbol{\eta})||p(\boldsymbol{\eta})]$. This term vanishes when the variational posterior is identical to the prior, i.e., $q(\boldsymbol{\eta}) = p(\boldsymbol{\eta})$. Similarly, the fourth and fifth terms represent the expected log-likelihood of the observed labels under the noise model, regularized by the entropy of the label posterior. These terms also cancel each other out under the assumption of a symmetric noise process, where $A$ is symmetric and $A = B$ in Eqs. (6) and (5). See Appendix A.4 for detailed derivation of the label noise terms cancellation under the symmetric noise process assumption. This condition is met in non-trivial noisy scenarios, for instance, with a uniform noise model where an error from any class is equally likely to transition to any other incorrect class (i.e., $\forall k, c \in \mathcal{Y}, k \neq c \Rightarrow A_{kc} = B_{kc} = 1/(|\mathcal{Y}| - 1)$).

In such a setting, maximizing the ELBO is equivalent to maximizing only the first term. As we saw, if we assume a noise-free setting, the posterior distribution $q(y_i^* | y_i, \eta_i)$ collapses to a point mass (i.e., a Kronecker

delta function centered at the observed label $y_i$), making the soft labels $q(y_{ik}^*|y_i)$ identical to the observed one-hot labels. Consequently, our objective function reduces precisely to the standard cross-entropy loss. Therefore, our method provides a principled generalization of the standard supervised learning framework, where the additional terms in the ELBO serve to regularize the posterior and enforce spatial correlation in the inferred label errors.

While the expectations over the discrete labels $y_i^*$ are tractable, the terms involving the continuous variable $\boldsymbol{\eta}$ pose a significant computational challenge. The number of elements in $\boldsymbol{\eta}$ equals the number of image pixels ($HW$), which can be in the hundreds of thousands, and these elements are spatially correlated. For instance, the negative KL divergence term, $-\text{KL}[q(\boldsymbol{\eta})||p(\boldsymbol{\eta})]$, is analytically computable since both $p(\boldsymbol{\eta}) = \mathcal{N}(\boldsymbol{\eta}|\boldsymbol{\mu}, \boldsymbol{\Sigma})$ and $q(\boldsymbol{\eta}) = \mathcal{N}(\boldsymbol{\eta}|\boldsymbol{m}, \boldsymbol{\Gamma})$ are Gaussians. Its value is given by:

$$-\text{KL}[q(\boldsymbol{\eta})||p(\boldsymbol{\eta})] = \frac{1}{2}\left(\log\frac{|\boldsymbol{\Gamma}|}{|\boldsymbol{\Sigma}|} + HW - \text{Tr}\left(\boldsymbol{\Sigma}^{-1}\boldsymbol{\Gamma}\right) - (\boldsymbol{m}-\boldsymbol{\mu})^T\boldsymbol{\Sigma}^{-1}(\boldsymbol{m}-\boldsymbol{\mu})\right). \tag{9}$$

However, evaluating this term is computationally prohibitive for large images, as it requires computing the determinants and inverses of the $HW \times HW$ covariance matrices, in addition to the trace and quadratic form terms. These challenges are efficiently addressed by leveraging the properties of the KMS-structured covariance, as we describe next.

## 4.2 KMS-structured covariance

An $n \times n$ KMS matrix $R_\rho$ is a type of symmetric Toeplitz matrix, where each element is defined using the parameter $\rho \in (-1, 1)$. Its inverse, $R_\rho^{-1}$, is a tridiagonal matrix. The explicit forms of $R_\rho$ and $R_\rho^{-1}$ are given by:

$$R_\rho := \begin{pmatrix} 1 & \rho & \rho^2 & \cdots & \rho^{n-1} \\ \rho & 1 & \rho & \cdots & \rho^{n-2} \\ \rho^2 & \rho & 1 & \cdots & \rho^{n-3} \\ \vdots & \vdots & \vdots & \ddots & \vdots \\ \rho^{n-1} & \rho^{n-2} & \rho^{n-3} & \cdots & 1 \end{pmatrix}, \quad R_\rho^{-1} = \frac{1}{1-\rho^2}\begin{pmatrix} 1 & -\rho & 0 & \cdots & 0 \\ -\rho & 1+\rho^2 & -\rho & \cdots & 0 \\ 0 & -\rho & 1+\rho^2 & \ddots & \vdots \\ \vdots & \vdots & \ddots & \ddots & -\rho \\ 0 & 0 & \cdots & -\rho & 1 \end{pmatrix},$$

and the determinant is computed as $|R_\rho| = (1 - \rho^2)^{n-1}$. See Appendix A.1 for mathematical derivations. While the KMS matrix is well-established in fields like signal processing for modeling AR(1) processes Grenander & Szegö (1958); Brockwell & Davis (1991), its unique properties — a sparse inverse and an analytic determinant — have been largely untapped as a computational tool in modern machine learning. Our work bridges this gap by demonstrating its utility as the key component for making the ECCD framework computationally tractable.

This matrix is useful for representing correlations among one-dimensional variables, where the correlation decays exponentially as the index distance between two variables increases regardless of their absolute positions.

Suppose a one-dimensional random variable $\mathbf{x}$ has a covariance structure given by

$$\mathbb{E}[(x_i - \mu_i)(x_j - \mu_j)] = \sigma_i\sigma_j\rho^{|i-j|}, \tag{10}$$

where $\mu_i$ is the mean of $x_i$. By utilizing the KMS matrix $R_\rho$, the covariance matrix can be expressed as

$$\boldsymbol{\Sigma} = VR_\rho V, \quad \boldsymbol{\Sigma}^{-1} = V^{-1}R_\rho^{-1}V^{-1}, \tag{11}$$

where $V$ is a diagonal matrix whose $(i, i)$-th element is $\sigma_i$.

In the case of two-dimensional spatial correlations, such as those in images, we utilize the Kronecker product. Suppose a two-dimensional random variable $\mathbf{x}$ has a covariance structure given by

$$\mathbb{E}[(x_{ij} - \mu_{ij})(x_{uv} - \mu_{uv})] = \sigma_{ij}\sigma_{uv}\rho^{|i-u|+|j-v|}, \tag{12}$$

where $\mu_{ij}$ is the mean of $x_{ij}$. Its matrix form can be expressed as

$$\mathbf{\Sigma} = V(R_\rho \otimes R'_\rho)V, \quad \mathbf{\Sigma}^{-1} = V^{-1}(R_\rho^{-1} \otimes R_\rho'^{-1})V^{-1}, \tag{13}$$

where we assume that the two-dimensional pixel index $(i, j)$ is mapped to a one-dimensional index $n$. $V$ is a diagonal matrix whose $(n, n)$-th element corresponds to $\sigma_{ij}$, and $\otimes$ denotes the Kronecker product.

Since $V$ is diagonal and $R_\rho^{-1}$ is tridiagonal, the inverse covariance $\mathbf{\Sigma}^{-1}$ is a sparse matrix. The properties of the KMS matrix provide an elegant solution to the computational challenges outlined in Section 4.1. First, the determinant of the covariance matrix can be computed analytically, bypassing the need for expensive numerical methods. Second, the sparse structure of the inverse matrix is crucial for the remaining terms. It reduces the complexity of computing the quadratic form and the trace term in the KL divergence (9) from $O((HW)^2)$ to $O(HW)$, which is linear in the number of pixels.

Beyond this computational speed-up, the sparse structure of the inverse also ensures that the quadratic form

$$(\boldsymbol{\eta} - \boldsymbol{\mu})^\top \mathbf{\Sigma}^{-1}(\boldsymbol{\eta} - \boldsymbol{\mu}) \tag{14}$$

involves only pairs of adjacent pixel variables. This, in turn, makes the conditional distribution $p(\eta_i \mid \boldsymbol{\eta}_{\setminus i})$ depend only on the neighboring pixels $\boldsymbol{\eta}_{\mathcal{N}_i}$, where $\boldsymbol{\eta}_{\setminus i}$ denotes all variables in $\boldsymbol{\eta}$ except $\eta_i$, and $\mathcal{N}_i$ represents the set of adjacent pixels of $i$.

In other words, the KMS-structured covariance ensures that the Gaussian distribution retains a Markov Random Field property while maintaining the tractability of both the inverse and the determinant of the covariance matrix. The effect of the correlation parameter $\rho$ is illustrated in Figure 1b. As shown, the model can confidently infer label errors when adjacent pixels are also likely to contain errors and the spatial correlation is strong (i.e., $\rho$ is high). This structure corresponds to the discretization of a continuous Gaussian Random Field with an exponential kernel under the $L_1$ distance. We provide a detailed theoretical analysis of these properties in Appendix B.

### 4.3 Optimization

We formulate the optimization problem as follows. Let $N$ be the number of training samples. We denote the model parameters by $\boldsymbol{\theta}$, the prior distribution parameters by $\boldsymbol{\omega} = \{\boldsymbol{\mu}, \mathbf{\Sigma}\}$, the image-specific variational posterior parameters by $\{\boldsymbol{\nu}^{(n)}\}_{n=1}^N = \{\boldsymbol{m}^{(n)}, \mathbf{\Gamma}^{(n)}\}_{n=1}^N$, and the negative ELBO for the $n$-th sample (from Eq.(7)) by $\mathcal{L}(\boldsymbol{\theta}, \boldsymbol{\omega}, \boldsymbol{\nu}^{(n)})$. The covariance matrices $\mathbf{\Sigma}$ and $\{\mathbf{\Gamma}^{(n)}\}_{n=1}^N$ are KMS-structured covariance matrices, each of which is parameterized by the diagonal elements and one spatial correlation parameter $\rho$ for KMS matrix. The spatial correlation parameter $\rho$ is initialized with the same value for the prior and posterior (see Section 5.2.3) and not optimized for the posterior either. Moreover, the label transition matrices $A$ and $B$ are not treated as learnable parameters: we assume a symmetric noise process in which $A$ is symmetric and $A = B$ in Eqs. (6) and (5). Under this symmetric noise assumption, the corresponding label noise terms in Eq. (7) cancel out and thus $A$ and $B$ do not explicitly appear in $\mathcal{L}(\boldsymbol{\theta}, \boldsymbol{\omega}, \boldsymbol{\nu}^{(n)})$.

Our training procedure, outlined in Algorithm 1, employs an alternating optimization scheme. For clarity, the algorithm is presented for a single sample per step, though this can be generalized to mini-batches. In each step for a given sample, the model parameters $\boldsymbol{\theta}$ are updated $K$ times, while the corresponding variational posterior parameters $\boldsymbol{\nu}^{(n)}$ are updated $M$ times. After training, the optimized model parameters $\boldsymbol{\theta}$ are used for inference on new images. Note that in Algorithm 1, the prior parameters $\boldsymbol{\omega}$ are treated as fixed, but they could also be optimized jointly with a sufficiently large training dataset. Thanks to the KMS structure, gradients for the KL divergence terms (involving log-determinants and traces of inverses) are computed fully analytically, avoiding high-variance Monte Carlo estimators for these regularization terms.

## 5 Experiments

### 5.1 Dataset

Research on segmentation with noisy labels has been especially active in both medical imaging and satellite remote sensing. In this work, we focus on two widely used benchmark datasets that represent these do-

---

**Algorithm 1** Training loop for alternating optimization of model and posterior parameters.

---

1: $\boldsymbol{\theta}, \boldsymbol{\omega}, \{\boldsymbol{\nu}_0^{(\ell)}\}_{\ell=1}^N \leftarrow$ Initialize parameters
2: **repeat**
3:     $n \leftarrow$ Randomly sample an index from $\{1, \ldots, N\}$
4:     **for** $k = 1, \ldots, K$ **do**
5:         Compute $\mathcal{L}(\boldsymbol{\theta}, \boldsymbol{\omega}, \boldsymbol{\nu}_0^{(n)})$ and its gradient w.r.t. $\boldsymbol{\theta}$
6:         $\boldsymbol{\theta} \leftarrow$ Update the model parameters using the gradient
7:     **end for**
8:     **for** $m = 0, 1, \ldots, M-1$ **do**
9:         Compute $\mathcal{L}(\boldsymbol{\theta}, \boldsymbol{\omega}, \boldsymbol{\nu}_m^{(n)})$ and its gradient w.r.t. $\boldsymbol{\nu}_m^{(n)}$
10:        $\boldsymbol{\nu}_{m+1}^{(n)} \leftarrow$ Update the posterior parameters using the gradient
11:     **end for**
12:     $\boldsymbol{\nu}_0^{(n)} \leftarrow \boldsymbol{\nu}_M^{(n)}$
13: **until** convergence of parameters $\boldsymbol{\theta}$

---

mains: the JSRT dataset for medical imaging and the WHU Building dataset for satellite imagery. Detailed descriptions of each dataset are provided below.

It is common practice in noisy-label segmentation research to artificially introduce label noise for evaluation because most publicly available datasets come with clean annotations Li et al. (2021); Song et al. (2022); Yao et al. (2023). Following this convention, we also employ synthetic noise in our experiments, with details of our noise generation process described in Section 5.2.1.

**JSRT Dataset** Shiraishi et al. (2000) is a publicly available dataset provided by the Japanese Society of Radiological Technology (JSRT), which comprises chest radiographs annotated with segmentation labels for the lung fields, heart, and clavicles. All images are of size $256 \times 256$ pixels, with 199 images designated for training and 50 for testing. Following the assumption common in noisy-label segmentation research—that no clean validation data is available—we evaluated our approach using the published split of 199 training images and 50 test images. Following a previous study Li et al. (2021), the Clavicle region was cropped to a fixed $96 \times 224$ area for training and inference.

**WHU Building Dataset** Ji et al. (2018) is a publicly available dataset for building segmentation containing satellite images of Christchurch with a resolution of $0.075\,\mathrm{m}$ covering an area of $450\,\mathrm{km}^2$. Manual annotations were provided for 22,000 buildings. The satellite images were downsampled to a resolution of $0.3\,\mathrm{m}$ and then divided into $512 \times 512$ tiles. Among the non-overlapping tiles, 4,736 were used for training, 1,036 for validation, and 2,416 for testing. Similar to JSRT, we adopted the published split for the training and testing data.

## 5.2 Experimental Setup

### 5.2.1 Noisy Label Settings.

In the JSRT dataset, we evaluated the impact of noise caused by boundary error, which is a critical issue in the medical imaging domain Yao et al. (2023); Zhang et al. (2020b). In accordance with previous works Xue et al. (2020); Zhang et al. (2020c); Xue et al. (2020); Li et al. (2021), we employed morphological transformations as synthetic noise. As illustrated in Figure 2, the noise settings consider not only boundary ambiguities (i.e., dilation or erosion) but also affine transformations. We examined the effect of noise using $\alpha$ to represent the proportion of affected data and $\beta$ to determine its strength, setting $\{(\alpha, \beta)\} = \{(0.3, 0.5), (0.5, 0.7), (0.7, 0.7)\}$.

In the WHU Building dataset, we modeled three types of label noise: omission noise, where labels are missing in regions that should be labeled Song et al. (2022); Liu et al. (2024); commission error, where labels are present in regions that should not be due to temporal changes Song et al. (2022); and boundary noise, which commonly occurs in segmentation tasks Song et al. (2022), as illustrated in Figure 2. We define the omission ratio $(\phi)$, the commission ratio $(\zeta)$, and the boundary noise probability $(\lambda)$ as the proportions of

Figure 2: **Examples of Noisy Labels.** In medical imaging, label noise is often simulated using morphological operations such as erosion, dilation, and affine transformations Li et al. (2021). Conversely, building segmentation commonly involves omission noise, commission errors, and boundary noise Song et al. (2022). The pink region indicates labels present in the noisy but absent in the clean label, while the blue region indicates the opposite.

the total instances in the image that are perturbed by noise. Specifically, the noise parameters are set as $\{(\phi, \zeta, \lambda)\} = \{(0.1, 0.0, 0.3), (0.2, 0.05, 0.5), (0.3, 0.1, 0.7)\}$.

### 5.2.2 Baselines.

We compare our method against several baselines, including cross-entropy loss (CE), noisy-label learning approaches for classification tasks (CoT Han et al. (2018), TriNet Zhang et al. (2020c), JoCoR Wei et al. (2020), EM Bekker & Goldberger (2016)), noise-robust losses for segmentation T-Loss Gonzalez-Jimenez et al. (2025), as well as techniques that account for partial spatial correlations, namely Superpixel (SP) Li et al. (2021) and SpatialCorrect (SC) Yao et al. (2023). For each method, we adopted the hyperparameters reported in the publicly available implementations or original papers, assuming them to be the best practices for their respective approaches. Although proposed method supports multi-label segmentation, many of the baseline methods are designed for binary segmentation. Therefore, in JSRT dataset, we conducted experiments by treating each class (Lung, Heart, Clavicle) as a separate binary segmentation task.

### 5.2.3 Implementation Details.

We use a U-Net Ronneberger et al. (2015) with an EfficientNet-B0 Tan & Le (2019) encoder, pre-trained on ImageNet Deng et al. (2009). The model is optimized using Adam Kingma & Ba (2014) with a learning rate of 0.001 and batch sizes of 16 (JSRT) and 32 (WHU Building). The parameters in the prior $p(\boldsymbol{\eta}) = \mathcal{N}(\boldsymbol{\eta}|\boldsymbol{\mu}, \boldsymbol{\Sigma})$ and the initial values of the parameters in the variational posterior $q(\boldsymbol{\eta}) = \mathcal{N}(\boldsymbol{\eta}|\boldsymbol{m}, \boldsymbol{\Gamma})$ are set as follows. These parameters (e.g., the hyperparameters of the noise-level prior) were tuned using a 15% validation subset of the WHU Building training set. Specifically, we performed a grid search over reasonable values and selected the setting that yielded the best validation performance (measured by validation ELBO). These hyperparameters were then fixed for all subsequent experiments across all datasets to evaluate the robustness of a generic setting. In practice, it is also possible to tune these hyperparameters for each new dataset if a validation set is available. The fixed values are: $\rho = 0.75$, $m = -5$, $\gamma = 1$, $\mu = -2$, and $\sigma = 1$.

### 5.3 Results for JSRT and WHU Building dataset

### 5.3.1 JSRT dataset.

As shown in the first left three columns of Figure 3, we report the Dice scores for each class (Lung, Heart, Clavicle) under all three noise conditions.

Under the moderate noise condition $(\alpha, \beta) = (0.3, 0.5)$, the CE baseline, which lacks explicit noise handling, experiences a noticeable drop of 6.2%, 2.4%, and 3.8% for each organ compared to learning with clean labels. On the other hand, most noise-robust methods exhibit only minor declines. Our proposed approach, in particular, shows a slight performance degradation in Lung (only 0.3%), indicating its strong robustness to mild noise.

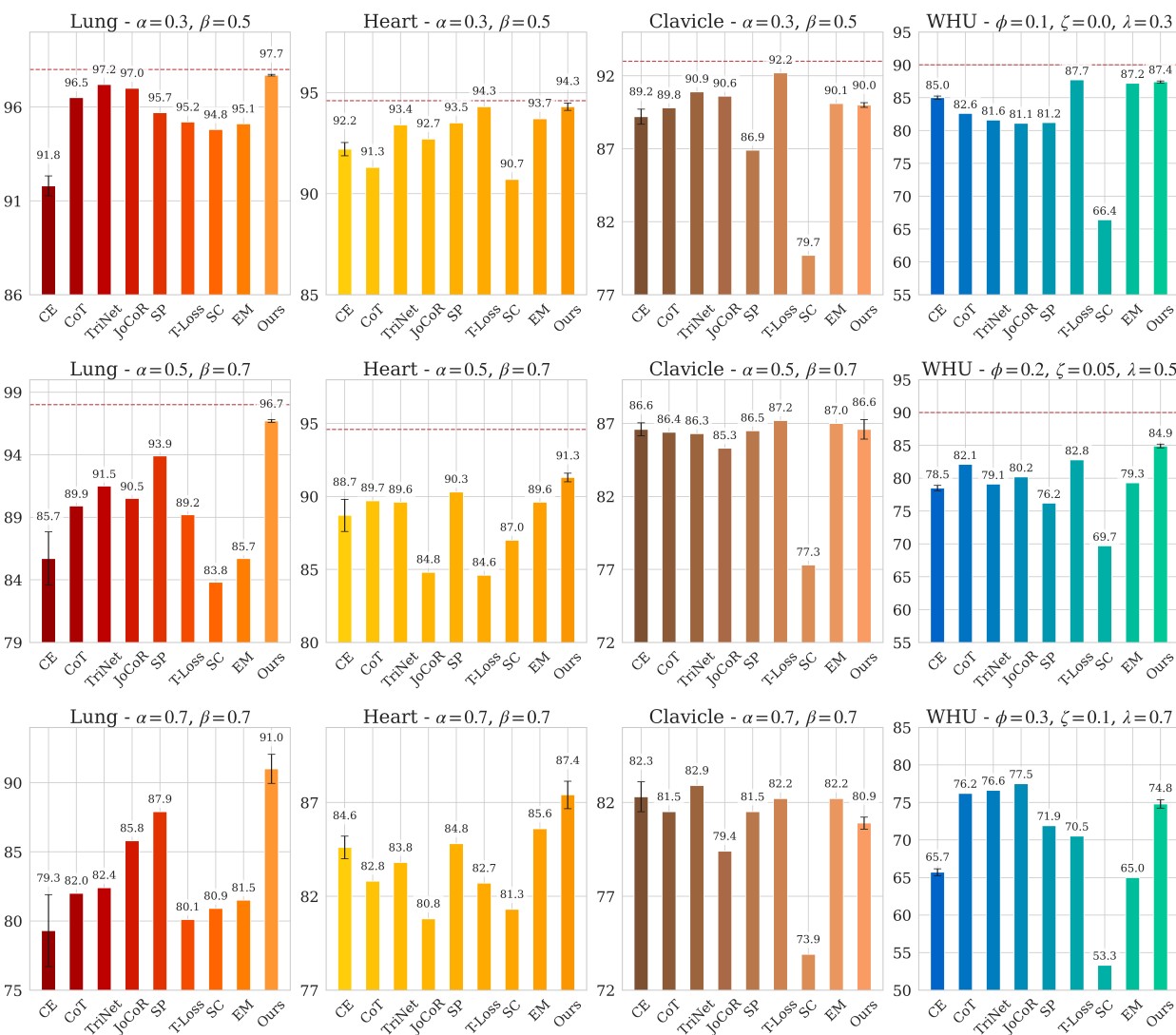

Figure 3: **Segmentation results on JSRT and WHU Building dataset.** The first left three columns show the segmentation results for three classes—Lung, Heart, and Clavicle—in the JSRT dataset, while the right column presents the binary segmentation results for WHU Building dataset in a bar chart. The rows correspond to three different noise intensity settings tested on each dataset, with Dice scores reported for JSRT and IoU scores for WHU Building dataset. The dotted lines in the bar charts represent the performance of a model trained with standard cross-entropy loss on clean labels, serving as an approximate upper bound.

As the noise level increases to $(\alpha, \beta) = (0.5, 0.7)$, CE undergoes a further 12.3% performance drop, and while other baselines also suffer more substantial declines, they remain relatively more stable than CE. Notably, our method continues to perform well on the Lung and Heart classes, demonstrating a limited reduction from the noise-free scenario. For the Clavicle class—an anatomically smaller and more challenging structure—both CE and some of the noise-robust methods experience larger drops, underscoring the sensitivity of thin or complex boundaries to label perturbations.

Under the heavier noise conditions $(\alpha, \beta) = (0.7, 0.7)$, all methods face considerable accuracy degradation. Even our approach, which remains competitive or superior for the Lung and Heart, shows more pronounced drops in Clavicle segmentation. This result highlights that extremely noisy annotations—particularly for

small or intricate regions—pose significant challenges. Nevertheless, compared to the other baselines, our method tends to retain higher Dice scores and lower variance in most cases.

One notable observation in our experiments is that the Clavicle segmentation task exhibits a larger performance degradation compared to the Lung and Heart classes under severe noise. A possible explanation is related to the EM-like feedback mechanism in ECCD. The method alternates between estimating the posterior over clean labels and updating the segmentation parameters $\theta$. When the current segmentation model is reasonably accurate, these two steps reinforce each other: reliable posterior estimates provide improved supervision, which in turn leads to better parameter updates.

However, under severe noise—especially for thin structures such as the clavicle—the initial segmentation model can be highly inaccurate. In this case, the inferred posterior over label errors may become unreliable, preventing the feedback loop from functioning effectively. As a result, the model may struggle to progressively correct label noise.

As a supplementary investigation, we additionally considered a simple warmup strategy, where $\theta$ is first pretrained using standard cross-entropy before optimizing the ECCD objective. This provides a better initialization and may help stabilize the early stage of training.

The results show that ECCD with warmup tends to yield slightly improved or comparable Dice scores for the clavicle segmentation under severe noise conditions. These observations are not conclusive, but are consistent with the hypothesis that the performance degradation may be related to optimization dynamics. Detailed results are provided in Appendix C.2.

To examine whether the proposed framework affects performance when labels are clean, we additionally conducted experiments on the JSRT dataset using the original clean annotations. The results show that ECCD achieves segmentation accuracy comparable to standard CE training. For example, on the lung, heart, and clavicle classes, the Dice scores of ECCD are within the confidence intervals of the CE baseline. These results suggest that the proposed framework does not degrade performance when label noise is absent. Detailed results are reported in Appendix C.3.

### 5.3.2 WHU Building dataset.

In the right column of Figure 3, we report IoU scores under three noise settings characterized by different omission, commission, and boundary probabilities $(\phi, \zeta, \lambda)$. For moderate noise $(\phi = 0.1, \zeta = 0.0, \lambda = 0.3)$ and $(\phi = 0.2, \zeta = 0.05, \lambda = 0.5)$, CE's performance declines by 5.0% and 11.5% from the upper bound, respectively. Methods like CoT, TriNet, JoCoR, and SP generally fall into the low- to mid-80% range. T-Loss and EM display a smaller gap from the clean-label upper bound, reflecting their robust design. Notably, our method reaches 87.4% for $(\phi = 0.1, \zeta = 0.0, \lambda = 0.3)$, just 2.6% below the upper bound, and continues to outperform other baselines at the next noise level.

When the noise becomes stronger $(\phi = 0.3, \zeta = 0.1, \lambda = 0.7)$, all methods exhibit further deterioration, although CoT, TriNet, JoCoR, and SP still maintain relatively higher IoU scores. Our method remains competitive, indicating its adaptability even under substantial label corruption.

The results on the JSRT and WHU Building datasets confirm that our method effectively handles various types of label noise. In our experimental setting, this gives SC a disadvantage, as it is designed specifically for boundary-related biases.

### 5.4 Effect of Spatial Correlation Parameter $\rho$

Figure 4 (a) shows how segmentation accuracy varies with the spatial correlation parameter $\rho$ for JSRT Lung, JSRT Heart, JSRT Clavicle, and WHU Building datasets. In the Lung, Heart, and WHU Building datasets, performance peaks around $\rho = 0.75$, suggesting that moderate correlation effectively captures spatial structure and improves segmentation. In contrast, both ignoring spatial dependencies $(\rho = 0)$ and assuming near-complete correlation $(\rho = 0.99)$ result in noticeable accuracy drops. Performance on the Clavicle class differs, remaining relatively stable from $\rho = 0$ to $0.75$, likely due to its smaller, elongated

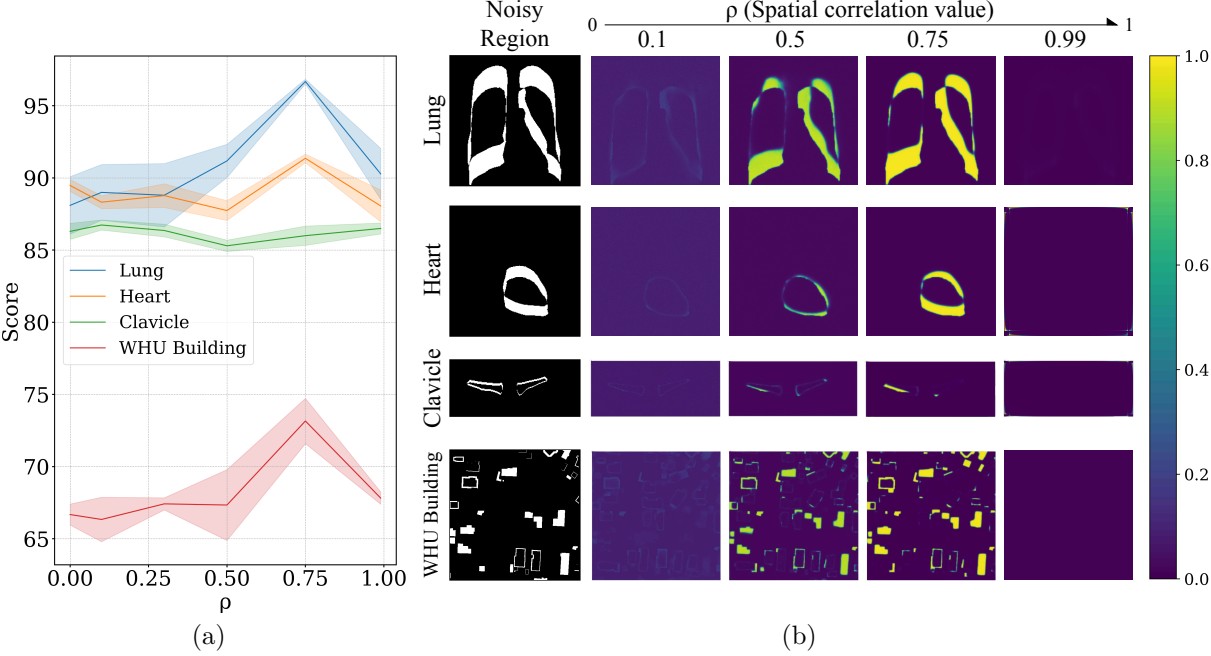

Figure 4: **(a) Effect of** $\rho$**.** Segmentation performance was evaluated across datasets based on $\rho$, which represents the degree of spatial correlation and affects segmentation performance across datasets. The results show that incorporating moderate spatial correlation enhances segmentation performance. Dice scores are reported for the Lung, Heart, and Clavicle, and IoU is used for the WHU Building dataset.
**(b) Estimated label error.** This figure visualizes the label error probability, computed as the sigmoid of $m$, the posterior mean of the logit of label error $\eta$. As shown in (a), higher accuracy in estimating $\eta$ leads to better segmentation performance.

shape, which reduces the benefits of spatial correlation. Figure 4 (b) further illustrates the estimated posterior label error probability in selected noisy regions under different $\rho$ values. At $\rho = 0.75$, mislabeled pixels are identified more accurately, indicating that the model's label-correction mechanism is most effective under moderate correlation. In contrast, pushing $\rho$ too high (e.g., 0.99) makes optimization unstable, leading to less reliable noise correction. Overall, these findings suggest that incorporating an appropriate level of spatial correlation significantly enhances the label correction mechanism.

## 5.5 Computational Overhead

Our method maintains image-specific variational parameters, which introduces additional computational overhead in both memory and wall-clock training time. We therefore quantify this overhead and compare against standard cross-entropy (CE) training and robust baselines. In our implementation, the main additional state is the posterior mean and variance of the latent Gaussian error field $\eta$, stored as two feature maps per training image. These maps are saved on disk and loaded on-the-fly with each mini-batch. As a result, persistent GPU memory usage is almost unchanged; the primary overhead is in computation.

To quantify this overhead, we measured per-epoch training time and GPU memory usage on the WHU building dataset with $128 \times 128$ crops (197 images) on an RTX A6000 (48GB). For the per-epoch training times in Table 1, SP[1] and SC[2] were measured using their official implementations. CoT, TriNet, and JoCoR were measured using the corresponding implementations included in the publicly available SP codebase.

---

[1]https://github.com/gaozhitong/SP_guided_Noisy_Label_Seg
[2]https://github.com/michaelofsbu/SpatialCorrection

Table 1: Training time per epoch and approximate additional GPU memory usage excluding image and mask tensors (WHU dataset, $128 \times 128$ crops, 197 images, RTX A6000 48GB). Here $\theta$ denotes the memory footprint of a single segmentation network, $B$ is the batch size, and $H, W$ are the image height and width, respectively. "Correlated ELBO (Full Cov.)" denotes the same ELBO optimization without exploiting the KMS covariance structure, which requires direct numerical computation of matrix inverses and determinants. $M$ corresponds to the number of gradient-based updates of the variational posterior in the E-step (Algorithm 1). OOM indicates GPU memory overflow.

| Method | $B = 1$ (s/epoch) | $B = 16$ (s/epoch) | Additional GPU memory usage (approx.) |
|---|---|---|---|
| CE | 86.68 | 2.63 | $\theta$ |
| CoT | 109.01 | 8.72 | $2\theta$ |
| TriNet | 118.47 | 10.50 | $3\theta$ |
| JoCoR | 102.13 | 8.55 | $2\theta$ |
| SP | 111.89 | 8.09 | $2\theta + BHW$ |
| T-Loss | 92.51 | 2.67 | $\theta$ |
| SC | 5.15 | 1.27 | $\theta$ |
| Pixel-wise EM | 88.34 | 2.93 | $\theta$ |
| Correlated ELBO (Full Cov., $M = 1$) | 2303.11 | OOM | $\theta + 2BHW$ (peak: $\theta + B(HW)^2$) |
| ECCD ($M = 1$) | 164.30 | 5.34 | $\theta + 2BHW$ (peak: $\theta + 9BHW$) |
| ECCD ($M = 10$) | 328.89 | 11.77 | $\theta + 2BHW$ (peak: $\theta + 9BHW$) |

Table 1 summarizes the results. With one variational update per iteration ($M = 1$), our method is roughly $2\times$ slower per epoch than standard CE training, and is of the same order as other robust baselines (e.g., Co-teaching, JoCoR). As the number of E-step update increases, the per-epoch training time increases. In our experiment, increasing $M$ from 1 to 10 increased the per-epoch training time from 164.30 to 328.89 s (batch size $B = 1$) and from 5.34 to 11.77 s (batch size $B = 16$). The dense covariance implementation becomes computationally infeasible even for moderate image sizes, highlighting the necessity of the KMS structure for scalable inference.

## 6 Conclusion

We have proposed a Bayesian framework for segmentation from noisy labels, introducing the logit of the label error probability as a continuous latent variable, $\boldsymbol{\eta}$. By incorporating the KMS matrix into the covariance structure of both the prior and the variational posterior distributions, our method efficiently computes the ELBO while capturing spatial correlations among adjacent pixels. This avoids the need to enumerate label combinations, improving computational efficiency. Our approach overcomes key limitations of conventional methods that assume pixel-wise independence or only partially model spatial correlation. Experiments on medical imaging and remote sensing datasets demonstrate superior robustness to noisy annotations when the spatial correlation parameter is set to a moderate value (e.g., $\rho = 0.75$). More broadly, ECCD provides a tractable framework for variational inference with spatially correlated discrete latent variables, which commonly arise in structured prediction problems beyond image segmentation. As such, ECCD offers a powerful tool for probabilistic modeling in domains where priors over correlated discrete variables are defined on structured domains such as spatial grids or temporal sequences. Potential avenues for future work include extending this framework to higher-dimensional spatial problems, like 3D segmentation and volumetric data analysis. Furthermore, as the underlying KMS matrix formulation is naturally suited for modeling exponentially decaying dependencies, the ECCD holds significant promise for domains beyond computer vision, particularly for modeling temporal correlations in time-series analysis and other sequential data.

Our method has some limitations. First, storing and optimizing posterior parameters for each training sample increases computational overhead; we quantify this overhead in Section 5.5. Second, spatial correlation may

be less beneficial when the target structures are small or elongated, as the spatial dependencies may not contribute effectively.

**Broader Impact Statement**

This work introduces the ECCD, a framework designed to enhance the robustness of machine learning models against spatially correlated label noise. We believe this research has the potential for significant positive societal impact, particularly in domains where high-quality data annotation is a major bottleneck. In medical imaging, our method could lower the barrier to developing diagnostic support tools by enabling the use of larger, more diverse, and imperfectly labeled datasets, potentially accelerating research and improving healthcare accessibility. Similarly, in remote sensing, the ECCD can facilitate more accurate and large-scale analysis of satellite imagery for applications such as environmental monitoring, disaster response, and urban planning, by effectively handling noise inherent in large-scale data collection.

At the same time, we recognize potential negative repercussions that users of this technology should consider. In high-stakes applications like medical diagnosis, over-reliance on the output of any automated system, including one trained with our method, can lead to automation bias and potential misdiagnosis if not used with expert oversight. We emphasize that this technology should be used as a decision-support tool to assist qualified professionals, not to replace them. Furthermore, like many remote sensing technologies, improved segmentation capabilities could be repurposed for surveillance or other applications with negative societal consequences. We advocate for the ethical use of this technology in accordance with established legal and ethical guidelines. Finally, the performance of our method relies on the assumption of stationary, exponentially decaying correlations in the label noise. Users should be aware that in scenarios where this assumption does not hold, the model's effectiveness may be reduced.

**Acknowledgments**

This research work was financially supported by the Ministry of Internal Affairs and Communications of Japan with a scheme of "Research and development of advanced technologies for a user-adaptive remote sensing data platform" (JPMI00316).

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

# Appendix

# A  Derivation details

In this section, we expand on several details that were only briefly mentioned in the main text. First, we provide a comprehensive explanation of the KMS matrix (see Section A.1). Next, we outline the intermediate steps in the derivation of the ELBO in our method and the derivation of the ELBO terms that account for spatial correlation (see Section A.2). We also derive the determinant and inverse of the covariance matrices $\Sigma$ and $\Gamma$, which are constructed using the KMS covariance (see Section A.3). In particular, Section A.3 details how we exploit the sparse structure of these inverses to achieve efficient computations.

## A.1  KMS Matrix

The *Kac–Murdock–Szegö (KMS) matrix* Fikioris (2018); Kac et al. (1953) is defined for a positive integer $n \in \mathbb{N}$ and a parameter $\rho \in (-1, 1)$ as

$$R_n(\rho) := \begin{bmatrix} 1 & \rho & \rho^2 & \rho^3 & \cdots & \rho^{n-1} \\ \rho & 1 & \rho & \rho^2 & \cdots & \rho^{n-2} \\ \rho^2 & \rho & 1 & \rho & \cdots & \rho^{n-3} \\ \rho^3 & \rho^2 & \rho & 1 & \ddots & \vdots \\ \vdots & \vdots & \vdots & \ddots & \ddots & \rho \\ \rho^{n-1} & \rho^{n-2} & \rho^{n-3} & \cdots & \rho & 1 \end{bmatrix} \in \mathbb{R}^{n \times n}. \tag{15}$$

The $(i, j)$ element of the KMS matrix is given by $\rho^{|i-j|}$, capturing the idea that the correlation between elements decays exponentially with the distance between indices.

**Determinant of** $R_n(\rho)$

To derive the determinant, we start with

$$\det R_n(\rho) = \begin{vmatrix} 1 & \rho & \rho^2 & \cdots & \rho^{n-1} \\ \rho & 1 & \rho & \cdots & \rho^{n-2} \\ \rho^2 & \rho & 1 & \cdots & \rho^{n-3} \\ \vdots & \vdots & \vdots & \ddots & \vdots \\ \rho^{n-1} & \rho^{n-2} & \rho^{n-3} & \cdots & 1 \end{vmatrix}. \tag{16}$$

For $i = 2, \ldots, n$, subtract $\rho^{i-1}$ times the first row from the $i$th row. This operation zeroes out the first column (except the first row) and introduces factors of $(1 - \rho^2)$ in the diagonal of the resulting submatrix. Proceeding recursively, one obtains

$$\det R_n(\rho) = (1 - \rho^2)^{n-1}. \tag{17}$$

This compact expression is key for efficient computation.

**Inverse of** $R_n(\rho)$

The inverse is given by

$$R_n(\rho)^{-1} = \frac{1}{\det R_n(\rho)} \operatorname{adj}(R_n(\rho)), \tag{18}$$

where the adjugate $\operatorname{adj}(R_n(\rho))$ is the transpose of the cofactor matrix. Through a series of elementary row operations and by exploiting the Toeplitz structure, it can be shown that the cofactors also share a patterned structure. In particular, one finds that

$$\operatorname{adj}(R_n(\rho)) = (1 - \rho^2)^{n-2} S_n, \tag{19}$$

with

$$S_n = \begin{bmatrix} 1 & -\rho & 0 & \cdots & 0 \\ -\rho & 1+\rho^2 & -\rho & \cdots & 0 \\ 0 & -\rho & 1+\rho^2 & \ddots & \vdots \\ \vdots & \vdots & \ddots & \ddots & -\rho \\ 0 & 0 & \cdots & -\rho & 1 \end{bmatrix}. \tag{20}$$

Thus, using $\det R_n(\rho) = (1 - \rho^2)^{n-1}$, we obtain

$$R_n(\rho)^{-1} = \frac{(1 - \rho^2)^{n-2}}{(1 - \rho^2)^{n-1}} S_n = \frac{1}{1 - \rho^2} S_n. \tag{21}$$

These derivations are essential in our framework for efficiently computing the covariance matrix inverse and determinant used in the Bayesian model.

## A.2 Derivation of ELBO (Eq. (7))

We assume the following generative model for our segmentation task:

$$p_{\boldsymbol{\theta}}(\mathbf{y}^*, \mathbf{y}, \boldsymbol{\eta} | \mathbf{x}) = p_{\boldsymbol{\theta}}(\mathbf{y}^* | \mathbf{x}) \, p(\mathbf{y} | \mathbf{y}^*, \boldsymbol{\eta}) \, p(\boldsymbol{\eta}) \tag{22}$$

$$= \left\{ \prod_i p_{\boldsymbol{\theta}}(y_i^* | \mathbf{x}) \right\} \left\{ \prod_j p(y_j | y_j^*, \eta_j) \right\} p(\boldsymbol{\eta}), \tag{23}$$

where the prior $p(\boldsymbol{\eta})$ is designed to capture spatial correlation (via a KMS covariance structure).

We introduce the variational distribution $q_\phi$ as

$$q_\phi := q(\boldsymbol{\eta}, \mathbf{y}^*|\mathbf{y}) = q(\boldsymbol{\eta}|\mathbf{y})\, q(\mathbf{y}^*|\boldsymbol{\eta}, \mathbf{y}) = q(\boldsymbol{\eta}) \prod_i q(y_i^*|\eta_i, y_i), \tag{24}$$

which approximates the true posterior $p_{\boldsymbol{\theta}}(\mathbf{y}^*, \boldsymbol{\eta}|\mathbf{y}, \mathbf{x})$. Then the marginal log-likelihood of the observed labels can be lower bounded as follows:

$$\log p_{\boldsymbol{\theta}}(\mathbf{y}|\mathbf{x}) = \log \int \sum_{\mathbf{y}^*} p_{\boldsymbol{\theta}}(\mathbf{y}|\boldsymbol{\eta}, \mathbf{y}^*, \mathbf{x})\, p_{\boldsymbol{\theta}}(\boldsymbol{\eta}, \mathbf{y}^*|\mathbf{x})\, d\boldsymbol{\eta} \tag{25}$$

$$= \log \mathbb{E}_{q_\phi} \left[ \frac{p_{\boldsymbol{\theta}}(\mathbf{y}|\boldsymbol{\eta}, \mathbf{y}^*, \mathbf{x})\, p_{\boldsymbol{\theta}}(\boldsymbol{\eta}, \mathbf{y}^*|\mathbf{x})}{q_\phi} \right] \tag{26}$$

$$\geq \mathbb{E}_{q_\phi} \left[ \log p_{\boldsymbol{\theta}}(\mathbf{y}|\boldsymbol{\eta}, \mathbf{y}^*, \mathbf{x}) \right] - KL\left[ q_\phi \,\|\, p_{\boldsymbol{\theta}}(\boldsymbol{\eta}, \mathbf{y}^*|\mathbf{x}) \right]. \tag{27}$$

We decompose the expectation term as:

$$\mathbb{E}_{q_\phi} \left[ \log p_{\boldsymbol{\theta}}(\mathbf{y}|\boldsymbol{\eta}, \mathbf{y}^*, \mathbf{x}) \right] \tag{28}$$

$$= \mathbb{E}_{q_\phi} \left[ \log p_{\boldsymbol{\theta}}(\mathbf{y}|\boldsymbol{\eta}, \mathbf{y}^*, \mathbf{x}) \right] \tag{29}$$

$$= \mathbb{E}_{q_\phi} \left[ \log p(\mathbf{y}|\boldsymbol{\eta}, \mathbf{y}^*) \right] \tag{30}$$

$$= \mathbb{E}_{q_\phi} \left[ \log \left( \prod_i p(y_i|\eta_i, y_i^*) \right) \right] \tag{31}$$

$$= \mathbb{E}_{q_\phi} \left[ \sum_i \log p(y_i|\eta_i, y_i^*) \right] \tag{32}$$

$$= \sum_i \mathbb{E}_{q_\phi} \left[ \log p(y_i|\eta_i, y_i^*) \right] \tag{33}$$

$$= \sum_i \mathbb{E}_{q(\eta_i, y_i^*|\mathbf{y})} \left[ \log p(y_i|\eta_i, y_i^*) \right] \quad (\text{marginalized by } (\boldsymbol{\eta}_{\backslash i}, \mathbf{y}_{\backslash i}^*)) \tag{34}$$

$$= \sum_i \mathbb{E}_{q(y_i^*|\eta_i, \mathbf{y})q(\eta_i)} \left[ \log p(y_i|\eta_i, y_i^*) \right] \tag{35}$$

$$= \sum_i \mathbb{E}_{q(y_i^*|\eta_i, y_i)q(\eta_i)} \left[ \log p(y_i|\eta_i, y_i^*) \right], \quad (y_i^* \perp\!\!\!\perp \mathbf{y}_{\backslash i}|\eta_i, y_i) \tag{36}$$

where $\backslash i$ denotes pixel indices other than $i$ on the image. We also decompose the negative KL divergence term as:

$$- KL \left[ q_{\boldsymbol{\phi}} \| p_{\boldsymbol{\theta}}(\boldsymbol{\eta}, \mathbf{y}^* | \mathbf{x}) \right] \tag{37}$$

$$= \mathbb{E}_{q_{\boldsymbol{\phi}}} \left[ \log \frac{p_{\boldsymbol{\theta}}(\boldsymbol{\eta}, \mathbf{y}^* | \mathbf{x})}{q_{\boldsymbol{\phi}}} \right] \tag{38}$$

$$= \mathbb{E}_{q_{\boldsymbol{\phi}}} \left[ \log p_{\boldsymbol{\theta}}(\boldsymbol{\eta}, \mathbf{y}^* | \mathbf{x}) - \log q_{\boldsymbol{\phi}} \right] \tag{39}$$

$$= \mathbb{E}_{q_{\boldsymbol{\phi}}} \left[ \log \underbrace{p_{\boldsymbol{\theta}}(\mathbf{y}^* | \boldsymbol{\eta}, \mathbf{x})}_{= \, p_{\boldsymbol{\theta}}(\mathbf{y}^* | \mathbf{x})} \underbrace{p_{\boldsymbol{\theta}}(\boldsymbol{\eta} | \mathbf{x})}_{= \, p(\boldsymbol{\eta})} - \log q(\mathbf{y}^* | \mathbf{y}, \boldsymbol{\eta}) \underbrace{q(\boldsymbol{\eta} | \mathbf{y})}_{= \, q(\boldsymbol{\eta})} \right] \tag{40}$$

$$= \mathbb{E}_{q_{\boldsymbol{\phi}}} \left[ \log \underbrace{p_{\boldsymbol{\theta}}(\mathbf{y}^* | \mathbf{x})}_{= \, \prod_i p_{\boldsymbol{\theta}}(y_i^* | \mathbf{x})} + \log p(\boldsymbol{\eta}) - \log \underbrace{q(\mathbf{y}^* | \mathbf{y}, \boldsymbol{\eta})}_{= \, \prod_i q(y_i^* | y_i, \eta_i)} - \log q(\boldsymbol{\eta}) \right] \tag{41}$$

$$= \mathbb{E}_{q_{\boldsymbol{\phi}}} \left[ \sum_i \log p_{\boldsymbol{\theta}}(y_i^* | \mathbf{x}) + \log p(\boldsymbol{\eta}) - \sum_i \log q(y_i^* | y_i, \eta_i) - \log q(\boldsymbol{\eta}) \right] \tag{42}$$

$$= \sum_i \mathbb{E}_{q(\eta_i) q(y_i^* | \eta_i, y_i)} \left[ \log p_{\boldsymbol{\theta}}(y_i^* | \mathbf{x}) \right] - \mathbb{E}_{q(\boldsymbol{\eta})} \left[ \log q(\boldsymbol{\eta}) \right] + \mathbb{E}_{q(\boldsymbol{\eta})} \left[ \log p(\boldsymbol{\eta}) \right] \tag{43}$$

$$- \sum_i \mathbb{E}_{q(\eta_i) q(y_i^* | \eta_i, y_i)} \left[ \log q(y_i^* | y_i, \eta_i) \right]. \tag{44}$$

Thus, the overall ELBO becomes

$$\log p_{\boldsymbol{\theta}}(\mathbf{y} | \mathbf{x}) \geq \sum_i \mathbb{E}_{q(\eta_i) \, q(y_i^* | y_i, \eta_i)} \left[ \log p_{\boldsymbol{\theta}}(y_i^* | \mathbf{x}) \right] - \mathbb{E}_{q(\boldsymbol{\eta})} \left[ \log q(\boldsymbol{\eta}) \right] + \mathbb{E}_{q(\boldsymbol{\eta})} \left[ \log p(\boldsymbol{\eta}) \right]$$

$$- \sum_i \mathbb{E}_{q(\eta_i) \, q(y_i^* | y_i, \eta_i)} \left[ \log q(y_i^* | y_i, \eta_i) \right] + \sum_i \mathbb{E}_{q(\eta_i) \, q(y_i^* | y_i, \eta_i)} \left[ \log p(y_i | y_i^*, \eta_i) \right]. \tag{45}$$

### A.3 Efficient ELBO Computation

In our framework, the spatially correlated prior $p(\boldsymbol{\eta})$ is modeled as a Gaussian distribution with a covariance matrix that incorporates spatial structure via a Kac–Murdock–Szegö (KMS) matrix:

$$\boldsymbol{\Sigma} = V \left( R_\rho \otimes R'_\rho \right) V, \tag{46}$$

where $R_\rho := R_H(\rho) \in \mathbb{R}^{H \times H}$ and $R'_\rho := R_W(\rho) \in \mathbb{R}^{W \times W}$ are the KMS matrices for the vertical and horizontal directions, respectively, and $V$ is a diagonal matrix (e.g., $V = \sigma I$ in a simplified setting).

Because the inverse of a KMS matrix is tridiagonal, the inverse of the Kronecker product $R_\rho^{-1} \otimes R_\rho'^{-1}$ is very sparse (with at most 9 nonzero entries per row). Hence, if $V = \sigma I$, then

$$\boldsymbol{\Sigma}^{-1} = V^{-1} \left( R_\rho^{-1} \otimes R_\rho'^{-1} \right) V^{-1} = \frac{1}{\sigma^2} \left( R_\rho^{-1} \otimes R_\rho'^{-1} \right). \tag{47}$$

Similarly, by exploiting the determinant properties of Kronecker products,

$$\det \boldsymbol{\Sigma} = (\det V)^2 \, (\det R_\rho)^W \, (\det R'_\rho)^H$$

$$= \left( \prod_{i=1}^{HW} \sigma_i \right)^2 (1 - \rho^2)^{W(H-1)} (1 - \rho^2)^{H(W-1)}. \tag{48}$$

Taking the logarithm yields

$$\log \det \boldsymbol{\Sigma} = 2 \sum_{i=1}^{HW} \log \sigma_i + W(H - 1) \log(1 - \rho^2) + H(W - 1) \log(1 - \rho^2). \tag{49}$$

The efficient computation of the trace terms in the ELBO is similarly achieved by noting that $\boldsymbol{\Sigma}^{-1}$ is sparse, which reduces the computational burden when evaluating expressions such as $\mathrm{tr}\left(\boldsymbol{\Gamma}\boldsymbol{\Sigma}^{-1}\right)$ and $\mathrm{tr}\left((\boldsymbol{\mu}-\boldsymbol{m})(\boldsymbol{\mu}-\boldsymbol{m})^T\boldsymbol{\Sigma}^{-1}\right)$.

### A.4 Derivation of Label Noise Terms Cancellation under Symmetric Noise Process Assumption

We consider the label noise terms, which are the last two terms of the ELBO in Eq. (7):

$$-\sum_i \mathbb{E}_{q(\eta_i)q(y_i^*|y_i,\eta_i)}\left[\log q(y_i^*|y_i,\eta_i)\right] + \sum_i \mathbb{E}_{q(\eta_i)q(y_i^*|y_i,\eta_i)}\left[\log p(y_i|y_i^*,\eta_i)\right] \tag{50}$$

$$=\sum_i \mathbb{E}_{q(\eta_i)q(y_i^*|y_i,\eta_i)}\left[\log p(y_i|y_i^*,\eta_i)\right] - \sum_i \mathbb{E}_{q(\eta_i)q(y_i^*|y_i,\eta_i)}\left[\log q(y_i^*|y_i,\eta_i)\right] \tag{51}$$

$$=\sum_i \mathbb{E}_{q(\eta_i)q(y_i^*|y_i,\eta_i)}\left[\log p(y_i|y_i^*,\eta_i) - \log q(y_i^*|y_i,\eta_i)\right] \tag{52}$$

$$=\sum_i \mathbb{E}_{q(\eta_i)}\left[\mathbb{E}_{q(y_i^*|y_i,\eta_i)}\left[\log p(y_i|y_i^*,\eta_i) - \log q(y_i^*|y_i,\eta_i)\right]\right] \tag{53}$$

The contribution of pixel $i$ for a given $\eta_i$ and an observed (noisy) label $o \in \mathcal{Y}$ can be written as

$$F_i(o,\eta_i) := \mathbb{E}_{q(y_i^*|y_{io}=1,\eta_i)}\left[\log p(y_{io}=1|y_i^*,\eta_i) - \log q(y_i^*|y_{io}=1,\eta_i)\right]. \tag{54}$$

It is sufficient to show $F_i(o,\eta_i) = 0$ for every $i,\eta_i,o$ under the symmetric noise process assumption. Expanding over $c \in \mathcal{Y}$, $F_i(o,\eta_i)$ can be expressed as follows:

$$F_i(o,\eta_i) = \sum_{c \in \mathcal{Y}} q(y_{ic}^*=1|y_{io}=1,\eta_i)\left[\log p(y_{io}=1|y_{ic}^*=1,\eta_i) - \log q(y_{ic}^*=1|y_{io}=1,\eta_i)\right] \tag{55}$$

$$=\sum_{\substack{c \in \mathcal{Y} \\ (c \neq o)}} q(y_{ic}^*=1|y_{io}=1,\eta_i)\left[\log p(y_{io}=1|y_{io}^*=1,\eta_i) - \log q(y_{io}^*=1|y_{io}=1,\eta_i)\right] \tag{56}$$

$$+ q(y_{io}^*=1|y_{io}=1,\eta_i)\left[\log p(y_{io}=1|y_{io}^*=1,\eta_i) - \log q(y_{ic}^*=1|y_{io}=1,\eta_i)\right] \tag{57}$$

$$=\sum_{\substack{c \in \mathcal{Y} \\ (c \neq o)}} r(\eta_i)A_{co}\left[\log(r(\eta_i)B_{oc}) - \log(r(\eta_i)A_{co})\right] + (1-r(\eta_i))\underbrace{\left[\log(1-r(\eta_i)) - \log(1-r(\eta_i))\right]}_{=0} \tag{58}$$

$$=\sum_{\substack{c \in \mathcal{Y} \\ (c \neq o)}} r(\eta_i)A_{co}\log\frac{r(\eta_i)B_{oc}}{r(\eta_i)A_{co}} \tag{59}$$

$$=\sum_{\substack{c \in \mathcal{Y} \\ (c \neq o)}} r(\eta_i)A_{co}\log\frac{B_{oc}}{A_{co}}. \tag{60}$$

Under the symmetric noise process assumption, in which $A$ is symmetric and $A = B$, we obtain

$$F_i(o,\eta_i) = \sum_{\substack{c \in \mathcal{Y} \\ (c \neq o)}} r(\eta_i)A_{co}\log\frac{B_{oc}}{A_{co}} = \sum_{\substack{c \in \mathcal{Y} \\ (c \neq o)}} r(\eta_i)A_{co}\log\frac{A_{oc}}{A_{co}} = \sum_{\substack{c \in \mathcal{Y} \\ (c \neq o)}} r(\eta_i)A_{co}\log\frac{A_{co}}{A_{co}} = 0. \tag{61}$$

Therefore,

$$\sum_i \mathbb{E}_{q(\eta_i)}\left[F_i(o,\eta_i)\right] = 0, \tag{62}$$

which shows that the last two terms of ELBO in Eq. (7) cancel each other out.

## B Theoretical Analysis and Discussions

## B.1 Marginal discrete distribution induced by the latent Gaussian field

Our model can be explicitly characterized as a **Logit-Gaussian Random Field** (or a Latent Gaussian Model with a logistic link function). Specifically, it models the log-odds (logits) of the label error probability as a spatially correlated Gaussian process, which is then mapped to discrete outcomes via a logistic/softmax function. From a modeling standpoint, the induced discrete distribution belongs to the same general construction as logit/probit models with a latent Gaussian field (often described as Gaussian-copula-type constructions in the statistical literature). While this general class of models is well-known, inference is typically computationally intractable, requiring expensive MCMC or approximations like INLA (Integrated Nested Laplace Approximation). Our specific contribution is identifying a subclass of these models that admits **efficient, closed-form variational inference**. By restricting the covariance of the latent field to the KMS structure, we enable the analytic computation of the ELBO (involving determinants and inverses) in $O(N)$ time, which is not possible for general Gaussian random fields. To the best of our knowledge, this specific KMS-based, ELBO-computable subclass has not been studied under a standard name. For this reason, we refer to it as an ELBO-Computable Correlated Discrete Distribution (ECCD) to highlight this key computational property. Regarding the dependency structure: the interactions are strictly pairwise in the latent Gaussian space (defined by the inverse covariance matrix $\Sigma^{-1}$). However, in the induced marginal discrete distribution $p(y|y^*)$ (after integrating out $\eta$), the dependencies do *not* generally reduce to simple pairwise interactions (like in a standard Ising or Potts model). Instead, the marginalization induces dense, higher-order dependencies among the discrete labels. We view this as an advantage, as it allows the model to capture complex, long-range error correlations (blobs) that are difficult to represent with simple pairwise discrete MRFs.

## B.2 Variational family, approximation bias, and gradients

**Approximation Bias:** We explicitly acknowledge the inherent approximation bias resulting from our choice of the variational family. Optimizing the ELBO is equivalent to minimizing the Kullback-Leibler (KL) divergence $KL(q(y^*|y,\eta)q(\eta)||p(y^*,\eta|y,x))$ between the joint variational posterior and the true joint posterior. Our variational formulation imposes specific structural constraints: 1) the label posterior is factorized as $q(y^*|y,\eta) = \prod_i q(y_i^*|y_i,\eta_i)$, ignoring direct dependence on $x$ and assuming the functional form of Eq. (5) with a fixed class transition matrix $A = B$ (shared across images) rather than optimizing $A$ per image; and 2) the latent field posterior is restricted to a Gaussian distribution $q(\eta) = \mathcal{N}(\eta|m,\Gamma)$ with KMS-structured covariance, where $(m,\Gamma)$ are optimized for each image. Since the true posterior over the latent field $\eta$ is generally non-Gaussian (and likely multimodal) due to conditioning on discrete labels via a non-linear likelihood, the optimal variational posterior within this constrained family does not converge to the true posterior. We accept this variational gap to achieve computational tractability and efficient inference.

**Gradients:** Regarding the optimization, a key advantage of our KMS-based framework is that the gradients with respect to the correlation parameters are computed fully analytically for the regularization terms. Specifically, the KL divergence term in the ELBO contains $\log \det \Sigma$ and $\mathrm{Tr}(\Sigma^{-1}\Gamma)$. Because we utilize the properties of KMS matrices to derive closed-form analytical expressions for the determinant and the sparse inverse $\Sigma^{-1}$ (as detailed in Appendix A.1), the gradients of these terms with respect to the correlation parameter $\rho$ are exact and do not require Monte Carlo estimators. (Note: The expected likelihood term involves an expectation over $q(\eta)$, which is estimated via the reparameterization trick, but the computationally heavy covariance-related terms are handled analytically.)

## B.3 Identifiability between segmentation parameters and noise correlations

We address the potential identifiability issues between the segmentation parameters $\theta$ and the noise correlation parameters:

1. **Structural separation: independence vs spatial correlation.** The clean labels are modeled by a standard discriminative segmentation network with conditional independence across pixels: $p_\theta(y^*|x) = \prod_{i=1}^{HW} p_\theta(y_i^*|x)$. Thus, conditioned on the image $x$, the random variables $y_i^*$ and $y_j^*$ are independent for $i \neq j$. The network may produce smooth probability maps (because CNNs tend to output smooth logits), but at the distributional level this is still a fully factorized model: it can control the **first-order moments** (per-pixel marginals), not the **second-order moments** (covariances) of the label field. By contrast, the noise model introduces a latent Gaussian field $\eta$ with KMS-structured covariance, and the resulting marginal distribution $p(y|y^*)$ exhibits non-zero spatial correlation between $y_i$ and $y_j$ (for nearby pixels) whenever $\rho > 0$. This correlation is stationary and controlled by the KMS-structured covariance; it is not something the factorized $p_\theta(y^*|x)$ can reproduce. Any systematic clustering of errors cannot be explained away by a different choice of $\theta$ in a factorized model; it must be attributed to the correlated label noise component.

2. **Practical behavior and remaining ambiguity.** In practice, identifiability is further helped by the fact that the segmentation model $p_\theta(y^*|x)$ learns image-dependent structure (e.g., anatomy) across the dataset, whereas the KMS noise model captures image-agnostic, stationary spatial clustering of residuals. Trying to explain correlated annotation noise purely via $\theta$ would require the network to memorize idiosyncratic "noise blobs", which generalizes poorly; the stationary noise model is a much more parsimonious fit. Empirically, across all our experiments we have **not** observed qualitatively different parameter configurations with essentially identical ELBO but very different predictions.

### B.4   Asymptotic and scaling behavior of the KMS approximation

**Convergence to Continuous Field:** The KMS covariance corresponds to a discretized exponential kernel, where the correlation decays exponentially with distance. By parameterizing $\rho = \exp(-\lambda h)$, this structure can be interpreted as a discrete approximation of a continuous Gaussian random field with an exponential covariance function. As the spatial resolution increases (i.e., grid spacing $h \to 0$) while scaling the correlation parameter as $\rho = e^{-\lambda h}$, the discrete field converges to a continuous Gaussian Random Field with a separable exponential covariance kernel defined under the $L_1$ distance: $k(\mathbf{u}, \mathbf{v}) \propto \exp(-\lambda\|\mathbf{u} - \mathbf{v}\|_1) = \exp(-\lambda|u_1 - v_1|)\exp(-\lambda|u_2 - v_2|)$, where $\mathbf{u}, \mathbf{v} \in \mathbb{R}^2$ are spatial coordinates. This implies that the inductive bias of our model is an anisotropic spatial correlation aligned with the coordinate axes.

**Boundary Conditions:** Regarding boundary conditions, it is important to clarify that the KMS (Toeplitz) matrix represents the exact covariance of a stationary process observed on a finite grid, **without imposing artificial boundary conditions**. Unlike spectral approximations that use Circulant matrices (which implicitly assume periodic boundaries, wrapping the image edges), the KMS structure correctly models a non-periodic field on a finite domain. Therefore, there are no implicit boundary artifacts at the image edges that would distort structures like clavicles. The primary limitation for thin, elongated structures (like clavicles) comes not from boundary conditions, but from the anisotropy of the $L_1$ metric: since the correlation decays based on Manhattan distance, the effective correlation length is direction-dependent (stronger along axes), which may not perfectly align with diagonal anatomical structures.

## C   Additional Experimental Results

### C.1   Noise-type-wise Results

### C.1.1   JSRT

We conduct experiments for three types of synthetic label perturbations—erosion, dilation, and affine transformations—each with different noise level settings ($(\alpha, \beta) = \{(0.3, 0.5), (0.5, 0.7), (0.7, 0.7)\}$) on JSRT Lung and Clavicle datasets.

We report noise-type-wise results on the JSRT Lung and Clavicle datasets in Tables 2. Across both structures, Dice scores tend to decrease as the noise level increases, indicating that stronger perturbations cause a

Table 2: Dice scores on JSRT dataset for each noise type and noise level.

| Noise type | $(\alpha, \beta)$ | Lung | | | | Clavicle | | | |
|---|---|---|---|---|---|---|---|---|---|
| | | CE | T-Loss | SP | Ours | CE | T-Loss | TriNet | Ours |
| Erosion noise | $(0.3, 0.5)$ | 89.0 | 88.4 | 94.9 | 92.2 | 87.9 | 89.2 | 88.7 | 88.0 |
| | $(0.5, 0.7)$ | 77.2 | 74.3 | 84.8 | 79.3 | 76.8 | 73.3 | 82.4 | 71.3 |
| | $(0.7, 0.7)$ | 66.5 | 58.1 | 72.5 | 48.9 | 67.8 | 72.4 | 76.6 | 56.3 |
| Dilation noise | $(0.3, 0.5)$ | 91.2 | 92.9 | 92.0 | 95.1 | 87.8 | 89.9 | 83.2 | 89.4 |
| | $(0.5, 0.7)$ | 79.8 | 81.0 | 82.3 | 82.2 | 83.2 | 83.1 | 76.4 | 83.9 |
| | $(0.7, 0.7)$ | 78.4 | 78.2 | 79.9 | 76.1 | 79.3 | 79.7 | 73.6 | 80.1 |
| Affine noise | $(0.3, 0.5)$ | 94.4 | 95.2 | 96.4 | 97.6 | 89.7 | 92.0 | 87.2 | 88.8 |
| | $(0.5, 0.7)$ | 89.6 | 87.7 | 93.5 | 94.0 | 80.7 | 78.5 | 81.3 | 82.2 |
| | $(0.7, 0.7)$ | 78.1 | 75.5 | 86.5 | 85.8 | 75.2 | 71.9 | 71.6 | 66.6 |

Table 3: IoU scores on WHU Building dataset with different noise types and noise levels.

| Noise type | Noise level | CE | T-Loss | Ours |
|---|---|---|---|---|
| Omission noise | 0.1 | 81.4 | 80.6 | 86.7 |
| | 0.3 | 65.4 | 62.8 | 77.3 |
| Commission noise | 0.1 | 82.9 | 85.5 | 87.4 |
| | 0.3 | 80.1 | 84.5 | 88.0 |

larger mismatch between the noisy supervision and the latent clean labels. The proposed method remains competitive under dilation and some affine perturbations, whereas its performance drops more substantially under erosion, particularly in the high-noise setting. One possible explanation is that erosion shrinks the foreground mask and may reduce the usefulness of spatially correlated cues for label correction, especially for thin structures such as the clavicle.

### C.1.2 WHU Building dataset

We conduct experiments for two types of synthetic label perturbations, omission noise and commission noise, with two noise levels each (omission ratio $\phi \in \{0.1, 0.3\}$, commission ratio $\zeta \in \{0.1, 0.3\}$) on the WHU Building dataset. Due to the large size of the original dataset, we use 30% of the data in this supplementary experiment.

Table 3 reports the IoU results. As shown in Table 3, our method achieved higher IoU than CE and T-Loss for both omission and commission noise at all tested noise levels.

### C.2 Supplementary Analysis of Warmup Initialization

In this section, we provide a supplementary analysis related to the discussion in Section 5.3.1 regarding the effect of initialization on the EM-like feedback mechanism in ECCD.

We conduct experiments on the JSRT dataset (Clavicle) under the severe noise setting $(\alpha, \beta) = (0.7, 0.7)$. We compare different initialization strategies for the segmentation parameters $\theta$: (i) no warmup (direct ECCD optimization from random initialization), and warmup using cross-entropy pretraining for (ii) 25, (iii) 50, and (iv) 100 epochs. Each setting is repeated three times, and we report the mean and standard deviation.

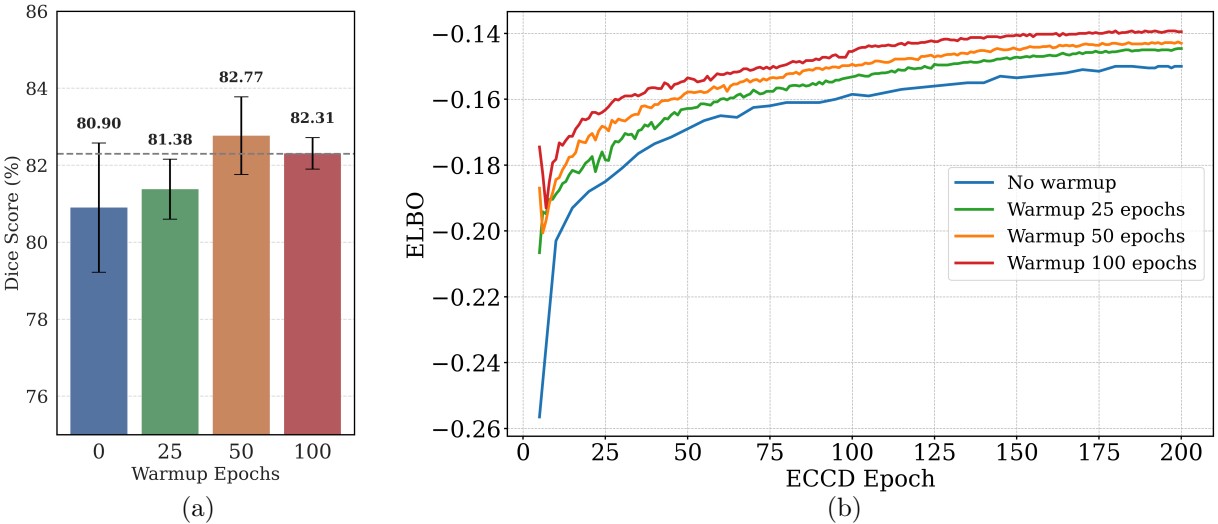

Figure 5: **(a) Final Dice scores for different warmup lengths (mean $\pm$ standard deviation). (b) ELBO during ECCD training for different warmup lengths.**

Table 4: Segmentation performance on the JSRT dataset with clean annotations. Mean Dice score and standard deviation are reported.

| Method | Lung | Heart | Clavicle |
|---|---|---|---|
| CE | $98.0 \pm 0.0$ | $94.9 \pm 0.0$ | $93.1 \pm 0.1$ |
| T-Loss | $98.1 \pm 0.0$ | $95.0 \pm 0.2$ | $93.9 \pm 0.1$ |
| Ours | $97.9 \pm 0.0$ | $94.8 \pm 0.1$ | $92.7 \pm 0.0$ |

Figure 5 (a) summarizes the final Dice scores, while Figure 5 (b) shows the evolution of the ELBO during training. We observe that longer warmup tends to result in higher ELBO values and more stable optimization. However, the differences in task performance, measured by Dice score, remain relatively small across settings.

Overall, these results are not conclusive, but are consistent with the hypothesis that initialization may influence the optimization dynamics of ECCD.

## C.3 Results with Clean Labels on JSRT

To evaluate whether the proposed method affects performance when labels are noise-free, we conducted additional experiments on the JSRT dataset using the original clean annotations. We compare the standard cross-entropy loss (CE), the robust loss T-Loss, and our ECCD method.

Table 4 reports the mean Dice scores over three runs together with the standard deviation. Overall, ECCD achieves performance comparable to CE across all three anatomical structures. The results indicate that the proposed framework does not significantly degrade segmentation performance when labels are clean.

