# OpenReview forum: "A Bayesian Approach to Segmentation with Noisy Labels via Spatially Correlated Distributions"
_TMLR — Accepted by TMLR_

### Review · Reviewer_SiDx · 2025-11-27

**Summary Of Contributions:**

The paper addresses semantic segmentation with noisy labels. The key insight is that annotation errors in real-world data (medical imaging, remote sensing) are not independently distributed across pixels, they exhibit spatial correlations. Adjacent pixels are more likely to share the same labeling errors.This paper proposes a Bayesian framework called ELBO-Computable Correlated Discrete Distribution (ECCD) for semantic segmentation with spatially correlated label noise. The key contributions are:

1. Novel probabilistic model: The authors introduce a continuous latent variable $\boldsymbol{\eta}$ representing the logit of label error probability, with spatial correlations encoded through a Gaussian prior with structured covariance.

2. Computational tractability via KMS matrices: By structuring the covariance matrix using the Kac-Murdock-Szegö (KMS) matrix, the authors achieve:
   - Closed-form determinant: $|\boldsymbol{\Sigma}| = \prod_i \sigma_i^2 \cdot (1-\rho^2)^{W(H-1)+H(W-1)}$
   - Sparse (tridiagonal) inverse enabling $O(HW)$ complexity for ELBO computation

3. Empirical validation: Experiments on JSRT (medical) and WHU Building (remote sensing) datasets demonstrate improvements over baselines, particularly at moderate noise levels.

Strengths:
- Elegant mathematical formulation connecting spatial correlation modeling to computational tractability
- Principled generalization of cross-entropy loss (reduces to CE when $q(\boldsymbol{\eta})=p(\boldsymbol{\eta})$ and noise-free)
- First application of KMS matrices for variational inference in this context

Weaknesses:
- The exponentially decaying isotropic correlation assumption may not match real noise patterns
- Per-sample variational parameters $\{\nu^{(n)}\}_{n=1}^N$ create memory overhead
- Limited analysis of when the method fails (e.g., clavicle results)

**Audience:**

Yes

**Audience Explanation:**

1. **Practical relevance:** Noisy labels are ubiquitous in medical imaging and remote sensing. Methods that explicitly model noise structure address a genuine bottleneck.

2. **Methodological novelty:** The introduction of KMS matrices for scalable variational inference is a technique that could transfer to other problems (time series, 3D data, sequential models).

3. **Theoretical interest:** The ECCD framework provides a principled probabilistic treatment that connects to classical graphical model theory (Markov Random Fields) while maintaining computational tractability.

4. **Reproducibility:** Code is provided, and the method uses standard architectures (U-Net + EfficientNet).

**Broader Impact Concerns:**

The authors provide a thoughtful broader impact statement addressing:
- Automation bias in medical diagnosis
- Potential for surveillance applications in remote sensing
- Limitations when model assumptions don't hold

**No additional concerns required.** The statement appropriately identifies risks and emphasizes the tool should assist, not replace, human experts. The acknowledgment of assumption-dependent performance is appropriate.

One minor addition could address: implications for annotation labor practices if methods that tolerate noise become standard, potentially reducing incentives for quality annotation infrastructure.

**Claims And Evidence:**

Yes

**Claims Explanation:**

1. **Mathematical derivations are correct:** The ELBO derivation (Appendix A.2) properly applies variational inference principles. The factorization assumptions are clearly stated and the marginalization over discrete $\mathbf{y}^*$ is handled correctly.

2. **KMS matrix properties are well-established:** The determinant and inverse formulas (Appendix A.1) are classical results that the authors correctly apply.

3. **Experimental improvements are demonstrated:** The proposed method outperforms baselines across most conditions, with particularly strong results for lung segmentation (97.7% Dice at moderate noise, only 0.3% below clean).

**Concerns:**

1. **Correlation structure mismatch:** The KMS structure assumes:
$$\text{Corr}(\eta_{ij}, \eta_{uv}) = \rho^{|i-u|+|j-v|}$$

This is an **isotropic, stationary, axis-aligned** correlation. Real label noise (e.g., from annotator disagreement on organ boundaries) likely follows the **object boundary geometry**, not a grid-aligned exponential decay. The paper does not analyze how well this assumption matches actual noise patterns.

2. **Missing ablation on correlation structure:** The authors vary $\rho$ (Figure 4a) but do not compare against alternative correlation structures (e.g., RBF kernel, boundary-aware correlations). This limits understanding of whether KMS specifically or spatial correlation generally drives improvements.

3. **Inconsistent clavicle results:** Performance on clavicle (thin, elongated structure) shows minimal benefit from spatial correlation. The authors acknowledge this but don't provide theoretical analysis of when their assumptions break down.

4. **Synthetic noise evaluation only:** All experiments use synthetically generated noise. While common in the literature, this means the claimed match between model assumptions and "real-world" noise patterns is not empirically validated.

5. **Statistical significance:** Error bars are shown but no formal statistical tests are reported. Some improvements appear within noise margins (e.g., WHU at highest noise level).

**Requested Changes:**

1. **Justify the KMS correlation structure for label noise:**

   The paper claims label errors exhibit spatial correlations but never validates that the specific KMS structure $\rho^{|i-u|+|j-v|}$ is appropriate. Please:
   - Empirically measure correlation patterns in real noisy annotations (e.g., from multi-annotator datasets like LIDC-IDRI)
   - Or provide theoretical justification for why exponentially decaying axis-aligned correlations model annotation errors
   - Or acknowledge this as a convenient approximation and discuss limitations

2. **Clarify the variational posterior structure:**

In Eq. (6), the conditional posterior $q(y_{ik}^* = 1 \mid y_{ic} = 1, \eta_i)$ depends explicitly on $\eta_i$, yet the paper also defines $q(\boldsymbol{\eta})$ as a Gaussian distribution. This creates a dependency structure: since $q(y_i^* \mid y_i, \eta_i)$ varies with $\eta_i$, the marginal $q(y_i^* \mid y_i)$ obtained by taking the expectation over $q(\eta_i)$ is generally \emph{not} a simple categorical distribution.

-- How is the expectation over $\eta_i$ computed in practice? Is it Monte Carlo sampling, numerical integration, or does a closed-form expression exist?

--  Algorithm~1 mentions gradients with respect to $\nu^{(n)}$ but does not specify what parameters are contained within $\nu^{(n)}$. Please clarify the definition and role of $\nu^{(n)}$ in the optimization.


3. **Address the computational overhead rigorously:**

   The paper claims overhead is "comparable to methods requiring multiple models" but:
   - How does memory scale with $N$ (training set size)?
   - What are wall-clock training times vs. baselines?
   - For the WHU dataset with $N=4736$ samples, each requiring posterior parameters $\{\mathbf{m}^{(n)}, \boldsymbol{\Gamma}^{(n)}\}$ where each is $512 \times 512 = 262,144$ dimensional, this is substantial.

4. **Provide statistical significance analysis:**

   Given the variance shown in error bars (e.g., Figure 3, WHU at high noise), please report p-values or confidence intervals for claims of improvement.

5. **Ablation on correlation structure:**

   Compare KMS against:
   - Independent noise ($\rho=0$) — already shown
   - Alternative kernels (e.g., Matérn, RBF approximated by low-rank)
   - Boundary-aware correlations (e.g., higher correlation along predicted edges)

6. **Analysis of failure cases:**

   The clavicle results show spatial correlation provides minimal benefit. Please analyze:
   - Is this because the structure is elongated (violating isotropy)?
   - Or because boundary noise dominates interior noise?
   - Provide guidance on when practitioners should expect the method to help.

7. **Extension to non-stationary correlations:**

   Discuss how the framework could accommodate spatially-varying $\rho(i,j)$, which would better model real annotation patterns where uncertainty varies (e.g., higher at boundaries).

8. **Clarify the connection to MRF/CRF literature:**

   The sparse inverse $\boldsymbol{\Sigma}^{-1}$ implies a Gaussian MRF structure. The paper mentions this briefly but could strengthen the theoretical contribution by connecting to classical results on Gaussian MRFs and explaining how ECCD differs from standard CRF approaches for segmentation.

---

> ### Author Response · Authors · 2025-12-19
>
> We sincerely thank the reviewer for the detailed and technically deep feedback. We appreciate that you found the probabilistic formulation of spatially correlated label noise and the KMS‑based construction interesting, and that you considered the segmentation experiments compelling. Your comments on the correlation structure and on the connection to MRF/CRF models helped us realize where we should clarify our modeling assumptions and contributions more explicitly.
>
> In response to your suggestions, we have been running several additional experiments
> (e.g., per–noise‑type analyses, clean‑label regimes, computational profiling, and
> multi‑seed runs for significance testing). Some of these experiments are still ongoing
> at the time of writing this rebuttal. In what follows, we report the results that are
> already available and explain how the remaining experiments will be incorporated into
> the revised manuscript.
>
> # 1. Correlation structure mismatch / justification for KMS / ablation / non‑stationary correlations
>
> Comments (paraphrased):
> > The KMS structure assumes isotropic, stationary, axis‑aligned correlation. Real label noise (e.g., annotator disagreement on organ boundaries) is likely boundary‑aligned, not grid‑aligned exponential decay. The paper does not analyze how well this assumption matches actual noise patterns.
>
> > The authors vary $\rho$ but do not compare against alternative correlation structures (e.g., RBF, boundary‑aware correlations). Please justify the specific KMS structure and discuss extensions to non‑stationary correlations.
>
> We fully agree that real annotation noise, especially around organ boundaries, is complex and cannot be captured exactly by any finite parametric model. Our goal is not to perfectly reproduce the true noise process, which is fundamentally unobservable, but to go beyond the standard pixel‑wise independent noise assumption in a way that remains computationally tractable.
>
> Existing label‑noise models for segmentation almost always assume independent per‑pixel noise; the proposed ECCD + KMS construction strictly generalizes these models by introducing spatially correlated errors while keeping the ELBO analytically computable. Even if the KMS prior is only an approximation, it captures the empirically observed tendency that “nearby pixels tend to share the same annotation error,” which i.i.d. models cannot express.
>
> Our covariance is defined not over pixel intensities but over the latent Gaussian field
> controlling the logits of label errors. Concretely, we use an RBF kernel with L1 distance
> on the grid, where $v_i \in \mathbb{Z}^2$ denotes the 2D spatial location of pixel $i$
> (and similarly for $v_j$), and $\lVert  \cdot \rVert_1$ is the L1 (Manhattan) norm:
> \begin{eqnarray}
>     \mathrm{Cov}(\eta_i, \eta_j) = \rho^{\lVert v_i - v_j \rVert_1},
> \end{eqnarray}
> which is the natural 2D extension of the 1D Kac–Murdock–Szegő (KMS) structure.
>  Similar stationary, exponentially decaying covariances (typically with L2 distance) have been used as Gaussian process priors for images and spatial fields, for example in Bayesian image super‑resolution (Tipping and Bishop, NIPS 2002)
> and in standard GP models (e.g., Rasmussen and Williams, 2006). We will add these references and clarify this connection in the revision.
>
> We also note that directly measuring the correlation structure of true label noise in real
> datasets is inherently difficult, because the clean labels $y^* $ are unknown by definition.
> Multi-annotator datasets can provide disagreement patterns as a useful proxy, but they do
> not reveal the underlying noise process relative to $y^* $ itself; in this work we therefore
> follow the common practice of studying synthetic but structured noise processes and treat
> the KMS prior as a computationally motivated approximation.
>
>
> The choice of L1 distance is driven by tractability: with L1, the covariance is KMS‑structured on the grid, so its inverse and log‑determinant can be handled analytically and the ELBO remains closed‑form. Using other distance metrics or more general kernels (e.g., Mat\'{e}rn, Euclidean RBF, explicit boundary‑aware kernels) would break the KMS structure and destroy this tractability, forcing us back to generic approximate inference for discrete MRFs/CRFs.

---

> > ### Author Response · Authors · 2025-12-19
> >
> > We also note that we employ the KMS structure both for the **prior** and for the
> > **variational posterior** over the latent error field. At the prior level, i.e., before
> > conditioning on any particular image, assuming stationarity and isotropy is not
> > unreasonable, since it simply encodes a generic preference for locally coherent error
> > patterns. The capacity of this prior is governed by the KMS correlation parameter
> > $\rho$: when $\rho = 0$, the KMS covariance reduces to a diagonal matrix, so the latent
> > error field becomes i.i.d.\ and the model collapses exactly to the standard
> > independent–noise case, while for $\rho > 0$ we gain the ability to represent spatially
> > correlated errors. In this sense, using a KMS–structured covariance does not reduce
> > expressive power relative to i.i.d.\ noise models; it only extends them to the correlated
> > setting.
> >
> > In our experiments we fix the correlation parameter in the **prior** covariance of
> > $\eta$ to a single value for simplicity, but in general this hyperparameter could be
> > chosen or tuned to match the expected strength and range of spatial correlations in a
> > given dataset. By contrast, the covariance of the **variational posterior**
> > $q(\eta)$ is also parameterized in KMS form with its own correlation parameter, which
> > is optimized from data. When the observations do not support a uniform spatial
> > correlation structure, this posterior correlation parameter tends to be driven close to
> > zero, so that the inferred latent error field effectively becomes i.i.d. even if the
> > prior uses a nonzero $\rho$.
> >
> > Regarding the relationship to explicitly boundary-aware structures, one could try to
> > encode such correlations directly in the clean labels $y^* $. For example, one may
> > introduce a latent Gaussian random field for each class $c \in \mathcal{Y}$, collect
> > them as  $\mathbf{\eta}  = (\mathbf{\eta}_1,  \cdots,  \mathbf{\eta}_Y )$, and define
> >
> > $$
> > p(\boldsymbol{\eta})
> > = \prod_{c \in \mathcal{Y}} \mathcal{N}(\boldsymbol{\eta}_c \mid \boldsymbol{\mu}_c, \Sigma),
> > $$
> >
> > $$
> > p(y^ * ) = \int p(y^ *  \mid \mathbf{\eta}) p( \mathbf{\eta})  d \mathbf {\eta}.
> > $$
> >
> > where $\Sigma$ has Toeplitz--KMS structure on the image grid. A simple choice is to
> > let
> > $p(y^ *  \mid \boldsymbol{\eta})$
> > be defined via a per-pixel softmax over the class‑wise
> > fields, i.e., for each pixel $ i $,
> >
> > $$
> >   p(y^*_i = c \mid \mathbf{\eta}) = \mathrm{softmax}_c\bigl(\boldsymbol{\eta}(i)\bigr),
> > $$
> >
> > where $\boldsymbol{\eta}(i)$ denotes the vector of logits $(\eta_{1i},\ldots,\eta_{|\mathcal{Y}|i})$
> > at pixel $i$. Because each class‑specific field $\boldsymbol{\eta}_c$ is spatially correlated
> > under the KMS prior, this construction induces a prior over $y^*$ that encourages pixels
> > with the same class label to cluster spatially, so that class boundaries are kept as small
> > as possible. This kind of behaviour is illustrated qualitatively in Figure~1(b) of the
> > paper, where pixels of the same class tend to form coherent regions with relatively few
> > boundaries.
> >
> > However, in such a formulation one typically works with a generative model of the form
> > $$
> > p(x, y) = \sum_{y^ * } p(x, y \mid y^ * )p(y^ * ),
> > $$
> > and then infers a posterior $q(y^* \mid x,y)$ in order to recover clean labels. The segmentation model $p(y^* \mid x)$ is then obtained only implicitly from this generative model; it is not parameterized and trained directly as a discriminative predictor
> > $p_\theta(y^* \mid x)$. As has been discussed extensively in the classical literature on generative vs. discriminative modeling, learning a generative model and then extracting a discriminative predictor from it often leads to suboptimal performance when the final goal is accurate prediction of $p(y^* \mid x)$.
> >
> > In this work we therefore retain an explicitly parameterized discriminative
> > segmentation network $p_\theta(y^* \mid x)$, and introduce the Toeplitz--KMS structure
> > only in the latent Gaussian error field that governs the label noise. This allows us to
> > (i) keep the segmentation model in the discriminative regime, while (ii) still exploiting
> > a tractable, spatially correlated noise model via the KMS structure at the latent level,
> > which in turn yields a closed‑form ELBO for ECCD. As discussed above, the same KMS‑based
> > latent field construction could in principle also be used to define boundary‑aware priors
> > over the clean labels $y^*$ (as in the toy example of Figure 1(b)), but systematically
> > exploring such label‑prior models lies beyond the scope of the present work.

---

> ### Author Response · Authors · 2025-12-19
>
> # 2. Connection to MRF / CRF literature and what is new about using KMS
>
>
> Comments (paraphrased):
> > Please clarify the connection to classical MRF / CRF models. What is actually new in using a KMS structure here? How does this relate to Gaussian MRFs and standard CRF‑based segmentation?'
>
> We appreciate this question and agree that we should more clearly position our work relative to classical MRF / CRF models.
>
> In traditional Gaussian MRFs for images, one often places a Gaussian prior $p(x) = N(x \mid m, V)$ directly on the continuous pixel intensities $x$, with $V^{-1}$ encoding local smoothness (e.g., via a graph Laplacian). Combined with a linear Gaussian observation model $p(y \mid x) = N(y \mid Wx, \sigma^2 I)$, the posterior $p(x \mid y)$ is Gaussian with closed‑form mean and covariance, but explicitly forming or inverting the covariance is infeasible in high dimensions. In practice, one solves linear systems involving $V^{-1}$
>  with iterative solvers or resorts to MAP estimation. Introducing a Toeplitz–KMS structure in $V$ can help here by making matrix–vector products and log‑determinants more tractable, and by enabling evidence maximization, but this is all in the space of continuous fields.
>
> Our setting is different in two key ways:
>
> 1. Discrete correlated variables via a latent Gaussian field.
> We are not modeling the pixel intensities or the labels themselves as a Gaussian field; instead, we introduce a latent Gaussian error field $\eta$ with KMS covariance and use it to induce a correlated distribution over discrete label errors. Classical conditional MRF models for segmentation directly place an conditional MRF on the labels $y$ (or on their continuous relaxations), which leads to a partition function
> \begin{align}
>     Z = \sum_{y_1}\cdots\sum_{y_N} \exp(-E(y_1, \cdots, y_N))
> \end{align}
> that is intractable in general, even for pairwise binary models such as Boltzmann machines. In contrast, our construction uses the KMS structure at the latent Gaussian level, so that the ELBO for the resulting ECCD is analytically computable, avoiding the need to approximate or bound the discrete partition function directly.
>
> 2. Noise modeling vs. label modeling.
> Classical CRF‑based segmentation models typically encode correlations between labels $y$ themselves (e.g., Potts models, contrast‑sensitive CRFs) and are used primarily for MAP inference of $y$ given observed images. Our focus is on modeling spatial correlation in label errors, not only in the labels: we explicitly separate clean labels and noisy labels, and use ECCD to model the correlated noise process, enabling Bayesian learning of a segmentation network under noisy annotations. This is, to our knowledge, a different use of MRF‑style structure than standard CRFs.
>
> We will add a dedicated paragraph in the revised manuscript to:
>
> - explain that Toeplitz–KMS structures have long been used as computationally convenient covariances for continuous Gaussian MRFs,
>
> - clarify that our novelty lies in using such a structure on a latent Gaussian error field to obtain a tractable correlated discrete distribution (ECCD) over label errors,
>
> - and contrast this with classical CRF models, where correlations are specified directly at the label level and inference typically relies on approximate MAP methods rather than an analytically tractable ELBO.
>
> We hope this clarifies both the role of the KMS structure in our work and how it differs from classical MRF/CRF formulations.

---

> ### Author Response · Authors · 2025-12-19
>
> # 3. Response to Requested Change 3 (computational overhead)
>
> Requested change (paraphrased)
> > Please quantify the computational overhead of the proposed method, including
> both time and memory, and compare against standard training.'
>
> Our implementation introduces only a small amount of additional state compared
> to standard cross-entropy (CE) training. Concretely, the main extra variables
> are the posterior mean and variance of the latent Gaussian error field
> $\eta$, which we store as two additional feature maps per training image.
> These statistics are \emph{not} kept permanently in GPU memory. Instead, they
> are stored on disk alongside the training images and loaded on-the-fly
> together with each mini-batch. As a result, the persistent GPU memory
> footprint is almost unchanged; the dominant overhead is in computation
> (extra forward/backward passes through the ECCD components and KMS-based
> Gaussian operations) rather than long-lived buffers.
>
> To quantify this overhead, we measured the per-epoch training time and GPU
> memory usage on the WHU building dataset with $128\times128$ crops
> ($197$ images) on an RTX A6000 (48GB). A summary of the results is provided
> in the table below \footnote{We will include this
> table, or an appropriately formatted version of it, in the revised
> manuscript.}
>
> | Method                     | batch size = 1 time (s/epoch) | batch size = 16 time (s/epoch) | Additional GPU memory besides image/mask (approx.)            |
> |----------------------------|-------------------------------|---------------------------------|----------------------------------------------------------------|
> | CE                         | 86.68                         | 2.63                            |  $\theta$                                                     |
> | CoT                        | 109.01                        | 8.72                            | $2 \theta$                                                   |
> | TriNet                     | 118.47                        | 10.5                            | $3 \theta$                                                   |
> | JoCoR                      | 102.13                        | 8.55                            | $2 \theta$                                                   |
> | SP                         | 111.89                        | 8.09                            | $2 \theta  + BHW$                                             |
> | T-Loss                     | 92.51                         | 2.67                            | $ \theta + \alpha$                                           |
> | SC                         | --                            | --                              | $\theta + \alpha$                                           |
> | EM                         | 88.34                         | 2.93                            | $\theta $                                                     |
> | Ours (E-step = 1)          | 164.30                        | 5.34                            | $ \theta + 2BHW$ (peak: $ \theta + 9BHW$)                   |
> | Ours (E-step = 10)         | 328.89                        | 11.77                           | $ \theta  + 2BHW$ (peak: $ \theta + 9BHW$)                   |
> | Ours (Dense, E-step = 1)   | 2303.11                       | GPU Out of Memory               | $ \theta  + 2BHW$ (peak: $\theta + B(HW)^2$)                |
>
>
> In particular:
> * For batch size $1$, the proposed method with $E$-step$=1$ is roughly
>         a factor of $\sim 2$ slower per epoch than standard CE training, and
>         its wall-clock time is of the same order as other robust training
>         baselines (e.g., Co-teaching, JoCoR).
>  * For batch size $16$, the same pattern holds: our method with
>         $E$-step$=1$ incurs about a $2\times$ increase in per-epoch time
>         relative to CE, while remaining comfortably within GPU memory.
>  * A naive dense-matrix implementation of the covariance (labeled
>         ``Ours Use Dense Matrix (E-step=1)'' in the table) runs into GPU
>         out-of-memory even at batch size $1$ on the same 48GB GPU, which
>         highlights the practical benefit of the KMS structure for scalability.
>
> We will report these measurements (including exact times and memory usage) in
> the revised manuscript, so that practitioners can clearly see the trade-off
> between robustness and computational cost.

---

> ### Author Response · Authors · 2025-12-19
>
> # 5. Response to Concern on analysis of failure cases
>
> Concern (paraphrased):
> > It would be helpful to understand in which regimes the proposed method can
> fail or even hurt performance.
>
> We agree that clarifying the failure modes of the method is important.
> Qualitatively, our approach has an EM-like structure: given a segmentation
> network $p_\theta(y^ *  \mid x)$ and a prior over label noise, we estimate the
> posterior distribution of the latent error field ($E$-step), and then use
> this posterior to update the segmentation parameters $\theta$ ($M$-step).
> When this process works well, there is a positive feedback loop:
>
> * a reasonably accurate segmentation model yields a good estimate of
>         the posterior over label errors, and
>  * this improved estimate of label errors in turn allows us to refine
>         the segmentation model.
>
> However, in very challenging regimes this positive feedback can fail to
> activate. For example, when the segmentation task is intrinsically very
> difficult or the label noise is extremely severe, the initial segmentation
> model may be too inaccurate to produce a meaningful estimate of the label
> error posterior. In such cases, the inferred noise pattern may not provide
> useful guidance for updating $\theta$, and the method may yield little or
> no improvement over standard CE training; in the worst case, performance can
> even degrade if the estimated error posterior is systematically biased.
>
> We will add a short discussion of these failure modes in the revised
> manuscript, emphasizing that the proposed method is most effective when there
> is at least a moderately good initial segmentation model (e.g., obtained by
> pre-training with CE on the noisy labels) so that the EM-like positive
> feedback between segmentation parameters and noise estimation can take effect.
>
> We thank the reviewer again for these insightful questions. We will incorporate the above clarifications, additional discussion, and explicit references into the revised manuscript.

---

> > ### Comment · Reviewer_SiDx · 2025-12-21
> >
> > We thank the authors for their detailed, technically thoughtful, and well-structured rebuttal. The responses clarify several key modeling choices, especially the motivation for using Toeplitz--KMS structure to retain ELBO tractability, the distinction from classical CRF-based segmentation, and the practical computational implications. The additional discussion on failure modes and discriminative vs. generative modeling is particularly helpful and improves the conceptual positioning of the work.
> >
> > That said, we believe a few precise points remain unclear.
> >
> >
> >
> > 1. Marginal form of the discrete posterior induced by the latent Gaussian field.
> >
> > The ECCD construction introduces a latent Gaussian field
> > $$
> > z \sim \mathcal{N}(\mu, \Sigma),
> > $$
> > with KMS covariance, which induces a correlated discrete distribution over label errors via a nonlinear (logistic or categorical) link. While the ELBO is computable in closed form, the marginal posterior over discrete noise variables requires integrating a nonlinear likelihood against a correlated Gaussian field.
> >
> > Question: Can the authors characterize the resulting discrete distribution more explicitly? For example, does ECCD correspond to a known class of models (e.g., a probit/logit Gaussian copula model), and under what conditions does the induced dependency structure reduce to pairwise interactions? A clearer characterization would help delineate the expressive limits of ECCD relative to classical correlated discrete models.
> >
> >
> >
> > 2. Consistency of the variational family and gradient estimation.
> >
> > Both the prior and the variational posterior over the latent field are assumed Gaussian with KMS-structured covariances, while the likelihood mapping the latent field to discrete labels is nonlinear.
> >
> > Question: Does the chosen variational family yield a consistent approximation in the sense that the optimal variational posterior converges to the true posterior as model capacity increases, or is there an inherent approximation bias due to the Gaussian assumption? In addition, are gradients with respect to the KMS correlation parameter computed fully analytically via closed-form expressions involving
> > $\log \det \Sigma \quad \text{and} \quad \Sigma^{-1},$
> > or do they rely on stochastic or Monte Carlo estimators?
> >
> >
> > 3. Identifiability between segmentation parameters and noise correlations.
> >
> > The model jointly learns a discriminative segmentation network
> > $p_\theta(y^* \mid x)$
> > and a correlated noise process parameterized through the latent field.
> >
> > Question: Are there identifiability issues between the segmentation parameters $\theta$ and the noise correlation parameters (e.g., structured errors being explained either by smoother predictions or by stronger noise correlations)? Have the authors observed regimes in which multiple parameter configurations achieve similar ELBO values but yield different predictive behavior, and if so, how is this ambiguity resolved in practice?
> >
> > 4. Asymptotic and scaling behavior of the KMS approximation.
> >
> > The tractability of ECCD relies on imposing a KMS structure on a finite image grid.
> >
> > Question: As spatial resolution increases, does the KMS-based construction converge, in any formal sense, to a continuous Gaussian random field with an exponential kernel under $L_1$ distance? Furthermore, how sensitive are the empirical results to the implicit boundary conditions induced by the Toeplitz assumption, particularly for thin or elongated structures such as clavicles?

---

> > > ### Author Response · Authors · 2026-01-06
> > >
> > > ### 1. Marginal discrete distribution induced by the latent Gaussian field
> > >
> > > Question:
> > > > ECCD introduces a latent Gaussian field $z \sim \mathcal{N}(\mu, \Sigma)$ with KMS covariance and a nonlinear link to discrete noise variables. Can the authors characterize the induced discrete distribution more explicitly? Does it correspond to a known model class (e.g., a Gaussian–copula construction), and under what conditions does the dependency structure reduce to pairwise interactions?
> > >
> > > Response:
> > > We thank the Reviewer for asking to clarify the theoretical positioning of our model relative to classical statistical models.
> > >
> > > Characterization and Relation to Known Models:
> > > Yes, the proposed model can be explicitly characterized as a **Logit-Gaussian Random Field** (or a Latent Gaussian Model with a logistic link function). Specifically, it models the log-odds (logits) of the label error probability as a spatially correlated Gaussian process, which is then mapped to discrete outcomes via a logistic/softmax function.
> > >
> > > From a modeling standpoint, the induced discrete distribution belongs to the same general construction as logit/probit models with a latent Gaussian field (often described as Gaussian-copula-type constructions in the statistical literature). While this general class of models is well-known, inference is typically computationally intractable, requiring expensive MCMC or approximations like INLA (Integrated Nested Laplace Approximation).
> > >
> > > Why "ECCD"?:
> > > Our specific contribution is identifying a subclass of these models that admits **efficient, closed-form variational inference**. By restricting the covariance of the latent field to the KMS structure, we enable the analytic computation of the ELBO (involving determinants and inverses) in $O(N)$ time, which is not possible for general Gaussian random fields. To the best of our knowledge, this specific KMS-based, ELBO-computable subclass has not been studied under a standard name. For this reason, we refer to it as an ELBO-Computable Correlated Discrete Distribution (ECCD) to highlight this key computational property.
> > >
> > > Dependency Structure:
> > > Regarding the dependency structure: the interactions are strictly pairwise in the latent Gaussian space (defined by the inverse covariance matrix $\Sigma^{-1}$). However, in the induced marginal discrete distribution $p(y|y^*)$ (after integrating out $\eta$), the dependencies do *not* generally reduce to simple pairwise interactions (like in a standard Ising or Potts model). Instead, the marginalization induces dense, higher-order dependencies among the discrete labels. We view this as an advantage, as it allows the model to capture complex, long-range error correlations (blobs) that are difficult to represent with simple pairwise discrete MRFs.

---

> > > ### Author Response · Authors · 2026-01-06
> > >
> > > ### 2.Variational family, approximation bias, and gradients w.r.t. KMS parametersQuestion:
> > >
> > > Question:
> > > > Does the chosen variational family yield a consistent approximation in the sense that the optimal variational posterior converges to the true posterior as model capacity increases, or is there an inherent approximation bias due to the Gaussian assumption? In addition, are gradients with respect to the KMS correlation parameter computed fully analytically via closed-form expressions involving $\log \det \Sigma$ and $\Sigma^{-1}$, or do they rely on stochastic or Monte Carlo estimators?
> > >
> > > Response:
> > >
> > > Approximation Bias: We explicitly acknowledge the inherent approximation bias resulting from our choice of the variational family. Optimizing the ELBO is equivalent to minimizing the Kullback-Leibler (KL) divergence $KL(q(y^* | y, \eta)q(\eta) || p(y^* , \eta | y, x))$ between the joint variational posterior and the true joint posterior.
> > > Our variational formulation imposes specific structural constraints: 1) the label posterior is factorized as $q(y^* | y, \eta)=\prod_i q(y^*_i | y_i, \eta _i)$, ignoring direct dependence on $x$ and assuming the functional form of Eq.~(5) with a fixed class transition matrix $A=B$ (shared across images) rather than optimizing $A$ per image. (Note: We have renamed the transition matrices $V$ and $W$ used in the original submission to $A$ and $B$ in this response and the revised manuscript. This change was made to avoid notation conflicts, as $W$ previously conflicted with the image width ($H \times W$) and $V$ conflicted with the diagonal matrix $V$ containing diagonal elements $\sigma_i$.) 2) the latent field posterior is restricted to a Gaussian distribution $q(\eta) = \mathcal{N}(\eta | m, \Gamma)$ with KMS-structured covariance, where $(m, \Gamma)$ are optimized for each image.
> > > Since the true posterior over the latent field $\eta$ is generally non-Gaussian (and likely multimodal) due to conditioning on discrete labels via a non-linear likelihood, the optimal variational posterior within this constrained family does not converge to the true posterior. We accept this variational gap to achieve computational tractability and efficient inference.
> > >
> > > Gradients: Regarding the optimization, a key advantage of our KMS-based framework is that the gradients with respect to the correlation parameters are computed fully analytically for the regularization terms.Specifically, the KL divergence term in the ELBO contains $\log \det \Sigma$ and $\text{Tr}(\Sigma^{-1}\Gamma)$. Because we utilize the properties of KMS matrices to derive closed-form analytical expressions for the determinant and the sparse inverse $\Sigma^{-1}$ (as detailed in Appendix A.1), the gradients of these terms with respect to the correlation parameter $\rho$ are exact and do not require Monte Carlo estimators. (Note: The expected likelihood term involves an expectation over $q(\eta)$, which is estimated via the reparameterization trick, but the computationally heavy covariance-related terms are handled analytically.)

---

> > > ### Author Response · Authors · 2026-01-06
> > >
> > > Question:
> > > > Are there identifiability issues between the segmentation parameters $\theta$ and the noise correlation parameters (e.g., structured errors being explained either by smoother predictions or by stronger noise correlations)? Have the authors observed regimes in which multiple parameter configurations achieve similar ELBO values but yield different predictive behavior, and if so, how is this ambiguity resolved in practice?
> > >
> > > Response: We appreciate this insightful question on identifiability.
> > >
> > > Our view is that:
> > > At the model-structure level, the segmentation and noise components play fundamentally different statistical roles, providing a form of identifiability.
> > > In practice, we have not observed qualitatively distinct solutions with similar ELBO but different predictions; remaining ambiguity is of the usual flat ELBO surface type.
> > >
> > > (1) Structural separation
> > > independence vs spatial correlationThe clean labels are modeled by a segmentation network with conditional independence: $p_\theta(y^* | x) = \prod_{i} p_\theta(y^*_i | x)$.
> > >
> > > Conditioned on $x$, variables $y_i^*$ are independent.  The network controls the first-order moments (marginals), not the second-order moments (covariances). Even if it produces smooth maps, it remains a factorized model. By contrast, the noise model introduces a latent Gaussian field $\eta$ with KMS covariance, $\eta \sim \mathcal{N}(\mu, \Sigma)$ where $\Sigma_{i,j} \propto \rho^{\| v_i -v_j \|_1}$.
> > >
> > > After marginalizing $\eta$, the resulting distribution $p(y | y^* )$ exhibits spatial correlations between nearby pixels $y_i, y_j$ whenever $\rho > 0$. This correlation is stationary and controlled by the KMS covariance; it is not something the factorized $p_\theta (y^* | x)$ can reproduce.Intuitively, even a smooth network cannot generate correlated residuals: it can only adjust per-pixel probabilities. Any systematic clustering of errors must be attributed to the correlated noise component, not $\theta$. This provides a strong separation between what $\theta$ and the noise parameters $\omega$ can explain.Formally, under the assumption that $\theta \mapsto p_\theta$ is injective, distinct parameter tuples ($\theta, \omega$) inducing the exact same joint distribution form a non-generic set. The practical concern is therefore near‑equivalence in ELBO rather than exact non-identifiability.
> > >
> > > (2) Practical behavior and remaining ambiguity
> > >
> > > In practice, identifiability is further helped by:
> > >
> > > - Image-dependent vs. Image-agnostic:  The segmentation model $p_\theta(y^* | x)$ learns image-dependent anatomy, whereas the KMS noise model captures image-agnostic, stationary spatial clustering. Explaining correlated noise via $\theta$ would require the network to memorize idiosyncratic “noise blobs”, which generalizes poorly; the stationary noise model is a much more parsimonious fit.
> > >
> > > - Spatial Stationarity:  The KMS structure implies that residual patterns across the dataset are highly informative about whether errors are scattered ($\rho \approx 0$) or clustered ($\rho > 0$).
> > >
> > > Empirically, we have not observed qualitatively different parameter configurations with identical ELBO but very different predictions. The remaining ambiguity is of the usual kind seen in deep neural networks (e.g., local optima), rather than a severe non-identifiability introduced by the noise model. We will clarify this structural separation and our empirical observations in the revision.

---

> > > ### Author Response · Authors · 2026-01-06
> > >
> > > ### 4. Asymptotic and scaling behavior of the KMS approximation
> > >
> > > Question:
> > > > As spatial resolution increases, does the KMS-based construction converge, in any formal sense, to a continuous Gaussian random field with an exponential kernel under $L_1$ distance? Furthermore, how sensitive are the empirical results to the implicit boundary conditions induced by the Toeplitz assumption, particularly for thin or elongated structures such as clavicles?
> > >
> > > Response:
> > >
> > > - Convergence to Continuous Field: Yes, your intuition is correct. The KMS-structured covariance matrix $R_{ij} = \rho^{ |i-j| }$ corresponds to the discretization of a continuous Ornstein-Uhlenbeck process. As the spatial resolution increases (i.e., grid spacing $h \to 0$) while scaling the correlation parameter as $\rho = e^{-\lambda h}$, the discrete field converges to a continuous Gaussian Random Field with a separable exponential covariance kernel defined under the $L_1$ distance:$$k(\mathbf{u}, \mathbf{v}) \propto \exp(-\lambda \|\mathbf{u} - \mathbf{v}\|_1) = \exp(-\lambda |u_1 - v_1|) \exp(-\lambda |u_2 - v_2|)$$ where $\mathbf{u}, \mathbf{v} \in \mathbb{R}^2$ are spatial coordinates. This implies that the inductive bias of our model is an anisotropic spatial correlation aligned with the coordinate axes.
> > >
> > > - Boundary Conditions: Regarding boundary conditions, it is important to clarify that the KMS (Toeplitz) matrix represents the exact covariance of a stationary process observed on a finite grid, **without imposing artificial boundary conditions**. Unlike spectral approximations that use Circulant matrices (which implicitly assume periodic boundaries, wrapping the image edges), the KMS structure correctly models a non-periodic field on a finite domain. Therefore, there are no implicit boundary artifacts at the image edges that would distort structures like clavicles. The primary limitation for thin, elongated structures (like clavicles) comes not from boundary conditions, but from the anisotropy of the $L_1$ metric: since the correlation decays based on Manhattan distance, the effective correlation length is direction-dependent (stronger along axes), which may not perfectly align with diagonal anatomical structures. However, as shown in our experiments, the model still provides robustness benefits compared to pixel-independent baselines.
> > >
> > >
> > >
> > > ### 5.Request for Revision Time regarding Additional Experiments
> > >
> > > Finally, we would like to address the recent feedback from Reviewer YuBG regarding the strength and consistency of our empirical evidence. Reviewer YuBG noted that our approach might underperform on smaller/thinner structures (e.g., clavicles) and in certain noise regimes, and suggested that a more explicit analysis is needed to understand these limitations.
> > >
> > > We agree that a comprehensive experimental picture is essential to validate the robustness of our method. To fully address these concerns and provide a convincing explanation for the behaviors observed on thin structures, we are currently re-evaluating our experimental design and plan to conduct additional targeted experiments. Therefore, we respectfully request a period of **several weeks** to complete these experiments and incorporate the new findings into the revised manuscript. We believe this additional time will allow us to significantly strengthen the paper's contribution and clarity.

---

### Review · Reviewer_855A · 2025-11-30

**Summary Of Contributions:**

This paper tackles semantic segmentation with noisy labels in settings such as medical imaging and remote sensing, where expert disagreement, misalignment and automatic labeling introduce structured errors. Instead of assuming label noise is pixel-wise independent, the authors explicitly model spatially correlated label errors and derive a Bayesian framework that generalizes standard cross-entropy training.

Summary of contributions:
1.They propose a new class of probabilistic models that explicitly capture spatial dependencies in correlated noisy labels for segmentation masks.

2.They make the resulting optimization problem computationally efficient and tractable by exploiting a Kac–Murdock–Szegö (KMS) matrix structure.


Strength of the paper:
1.The paper is well written, clearly motivates the limitations of standard cross-entropy under label noise, and provides solid mathematical derivations that show how their ELBO objective reduces to standard training as a special case.
2.The analysis of the spatial correlation parameter ρ (e.g., Figure 4) gives good intuition on how model assumptions relate to performance and helps interpret the behavior of the method.

Limitations:
1. The method requires storing and optimizing variational posterior parameters per training sample, which increases memory and computation compared to standard segmentation training or simpler noisy-label methods.

**Audience:**

Yes

**Audience Explanation:**

The problem addressed—robust segmentation under structured, spatially correlated label noise—is highly relevant in practical settings, especially in medical imaging and remote sensing, where imperfect and inconsistent annotations are the norm rather than the exception. Making the training procedure explicitly aware of such noise is an important direction for reliable deployment of segmentation models. The proposed framework and findings could therefore be of clear interest to TMLR’s audience, both for practitioners facing noisy supervision in real applications and for researchers interested in developing or extending methods for learning with noisy labels.

**Broader Impact Concerns:**

No concern

**Claims And Evidence:**

Yes

**Claims Explanation:**

The main claim—that explicitly modeling spatially correlated label noise leads to improved robustness compared to methods that assume independent noise—is supported by the experiments: their approach clearly outperforms baselines in high-noise regimes and achieves comparable performance under low-noise conditions, which is consistent with their hypothesis.

The second claim—regarding computational efficiency—is supported in a more limited sense. The use of the KMS matrix structure is convincingly shown to make the required linear algebra operations within their Bayesian framework more tractable and efficient than a naïve implementation. However, when compared to standard segmentation baselines, their overall method still appears computationally heavier, so this efficiency claim would benefit from clearer empirical comparisons (e.g., time and memory usage) against other methods.

**Requested Changes:**

Comments for securing my recommendation for acceptance:
- The proposed framework is designed for settings where the type and amount of label noise are unknown, and the model is not explicitly informed about the noise level. It would be very helpful to discuss how the method behaves in the opposite regime, i.e., when training on essentially clean data. If we train the model on completely clean labels (without injecting noise), does the performance degrade compared to standard training, and if so, by how much? Even if a full experimental study is beyond the scope of this paper, providing some intuition or empirical evidence for this scenario would be valuable, as many readers may wonder whether explicitly modeling label noise can harm performance when the actual noise level is very low or zero.

Comments to strengthen the work:
- 4.3 Optimization section:    In this section you write: “We denote the parameters of the prior distribution as \(\omega\), the image-specific parameters of the variational posterior as \(\{\nu^{(n)}\}_{n=1}^N\).”
    It would improve readability to clarify more explicitly what these parameters are and how they relate to the notation introduced earlier.

- Section 5.2.3 (Implementation Details).
You write: “These parameters were tuned using 15% of the WHU Building training set and validation sets, then fixed for all experiments.”
Please elaborate on the tuning procedure. For example:
Were these parameters treated as trainable variables jointly optimized with the network parameters, but only on that 15% subset?
Why are they not kept trainable during the full training on the entire dataset?
Is there an issue of optimization stability or overfitting that motivates this design choice?
Clarifying this would make the implementation details easier to reproduce and interpret.

- Figure 3.
The meaning of the dotted line is not clear, and it appears only in some subplots but not others. Please explain what the dotted line represents in the caption and, if relevant, why it is only present in some panels.
In addition, your method is shown with standard deviation bars while other methods are not. Please clarify in the caption (or main text) whether the baselines were also run with multiple seeds and, if so, why their variability is not shown.

- Figure 4.
In the left panel, for the Clavicle class, it appears that \(\rho = 1.0\) can yield better performance than the other \(\rho\) values, which seems somewhat at odds with the pattern observed for the other datasets and with the right panel. In the text you mention that, due to the small area, the result is not very dependent on \(\rho\), but I could not find any discussion of the specific behavior at \(\rho = 1.0\).
    I suggest adding a brief explanation of this case (e.g., why \(\rho = 1.0\) behaves differently here, or whether this is within expected variance) to better align the figure with the textual discussion.

---

> ### Author Response · Authors · 2025-12-21
>
> We sincerely thank the reviewer for the careful and thoughtful review.
> We are glad that you found the Bayesian formulation for spatially correlated label noise and the experimental evaluation on medical and remote sensing segmentation tasks interesting.
> Following your suggestions (especially on behaviour under clean labels and experimental details), we ran additional experiments:
> per–noise‑type results on JSRT lung and clavicle, behaviour under essentially clean labels on JSRT lung / heart / clavicle, and a systematic measurement of training time and memory usage.
> Some runs (e.g., SP for a few settings and WHU Building) are still in progress; below we report completed results and how they will be reflected in the revision.
>
> ### 1. Behaviour on essentially clean data
>
> **Comment (quoted):**
>
> > It would be very helpful to discuss how the method behaves in the opposite regime, i.e., when training on essentially clean data.
>
> We agree that understanding the clean‑label regime is important. When labels are clean, the optimal loss for training the segmentation network is standard cross‑entropy (CE), corresponding to maximum likelihood under a noise‑free model.
> By contrast, our method ECCD assumes that labels may contain spatially correlated noise, introduces a latent error field, and performs approximate Bayesian inference by maximizing an ELBO.
>
> When labels are perfectly clean, this extra modeling is not needed and can introduce small estimation errors due to (i) ELBO approximation and (ii) finite data; in the limit where the inferred noise level goes to zero, ECCD reduces to CE.
>
> We encode the a priori label noise via a latent Gaussian field
> $\boldsymbol{\eta} \sim \mathcal{N}(\mu \mathbf{1}, \Sigma)$,
> where $\mathbf{1}$ is the all‑ones vector and $\Sigma$ is KMS‑structured with
> $\Sigma_{ij} = \sigma^2 \rho^{\lVert v_i - v_j \rVert_1}$.
> Here $v_i \in \mathbb{Z}^2$ denotes the 2D location of pixel $i$ (and similarly for $v_j$), and $\lVert \cdot \rVert_1$ is the L1 (Manhattan) norm.
> In our experiments we use $\mu = -2$, $\sigma = 1$, and $\rho = 0.75$, expressing a prior belief that the noise level is small but nonzero while allowing spatially correlated fluctuations.
> Under this prior, we ran additional experiments on *clean* labels for JSRT lung, heart, and clavicle, using the same architecture and training protocol as in the main paper.
>
> We will summarise these clean‑label results in the revised manuscript as follows.
>
> **Table 1: Dice scores on JSRT Lung (clean labels, no synthetic noise).**
>
> | Method | mean    | std     | 95% CI               |
> |--------|---------|---------|----------------------|
> | CE     | 0.98021 | 0.00010 | [0.97996, 0.98046]   |
> | T-Loss | 0.98072 | 0.00021 | [0.98020, 0.98124]   |
> | Ours   | 0.97937 | 0.00037 | [0.97845, 0.98029]   |
>
> **Table 2: Dice scores on JSRT Heart (clean labels, no synthetic noise).**
>
> | Method | mean    | std     | 95% CI               |
> |--------|---------|---------|----------------------|
> | CE     | 0.94900 | 0.00030 | [0.94825, 0.94975]   |
> | T-Loss | 0.94985 | 0.00182 | [0.94533, 0.95437]   |
> | Ours   | 0.94832 | 0.00053 | [0.94700, 0.94964]   |
>
> **Table 3: Dice scores on JSRT Clavicle (clean labels, no synthetic noise).**
>
> | Method | mean    | std     | 95% CI               |
> |--------|---------|---------|----------------------|
> | CE     | 0.93105 | 0.00131 | [0.92779, 0.93431]   |
> | T-Loss | 0.93873 | 0.00064 | [0.93714, 0.94032]   |
> | Ours   | 0.92707 | 0.00032 | [0.92628, 0.92786]   |
>
> In these clean‑label experiments, ECCD performs slightly worse than CE on all three JSRT tasks, while remaining close in absolute Dice.
> This is exactly what we expect from the modeling perspective: when there is essentially no label noise to correct, CE is already statistically optimal for training the segmentation network, and ECCD cannot improve upon it—at best, when the inferred noise level goes to zero, ECCD recovers the CE solution.
> With finite data and approximate variational inference, ECCD can incur a small additional variance or approximation error, which explains the observed gap on clean labels.
> In the revised manuscript we will (i) add these clean‑data experiments and (ii) explicitly discuss this trade‑off between robustness under label noise (where ECCD provides clear gains) and the slight degradation that can occur when labels are perfectly clean.

---

> ### Author Response · Authors · 2025-12-21
>
> ### 2. Additional breakdown by noise type
>
> Although you did not explicitly request a per–noise-type breakdown, we found it useful to add such results to better characterize when the proposed method helps. Using the JSRT lung and clavicle datasets, we ran experiments for three types of synthetic label perturbations—erosion, dilation, and affine transformations—each with two parameter settings (e.g., $(a,b)=(0.3,0.5)$ and
> $(0.5,0.7)$ as in the attached slides).
>
> In the revised manuscript we will add tables of the following form for JSRT lung and clavicle, respectively:
>
> **Table 4: Dice scores on JSRT Lung for each noise type and noise level.**
> **Erode noise**
> |       | CE     | T-Loss | SP             | Ours   |
> |-------|--------|--------|----------------|--------|
> | $(a,b)=(0.3,0.5)$ | 0.88963| 0.88365| – (in progress)| 0.92242|
> | $(a,b)=(0.5,0.7)$  | 0.77221 | 0.74263| – (in progress)| 0.79265 |
>
> **Dilate noise**
> |       | CE     | T-Loss | SP             | Ours   |
> |-------|--------|--------|----------------|--------|
> | $(a,b)=(0.3,0.5)$ | 0.91176 | 0.92942 | – (in progress)| 0.9511 |
> | $(a,b)=(0.5,0.7)$  |0.79791 | 0.80962 | – (in progress)| 0.82223|
>
> **Affine noise**
> |       | CE     | T-Loss | SP             | Ours   |
> |-------|--------|--------|----------------|--------|
> | $(a,b)=(0.3,0.5)$ | 0.94411 | 0.95193| – (in progress)| 0.97639 |
> | $(a,b)=(0.5,0.7)$  |0.89626 | 0.87651| – (in progress)| 0.93977 |
>
>
> **Table 5: Dice scores on JSRT Clavicle for each noise type and noise level.**
> **Erode noise**
> |       | CE     | T-Loss | SP             | Ours   |
> |-------|--------|--------|----------------|--------|
> | $(a,b)=(0.3,0.5)$ | 0.87875| 0.89187| – (in progress)| 0.8804|
> | $(a,b)=(0.5,0.7)$  | 0.76835 | 0.73318| – (in progress)| 0.71267 |
>
> **Dilate noise**
> |       | CE     | T-Loss | SP             | Ours   |
> |-------|--------|--------|----------------|--------|
> | $(a,b)=(0.3,0.5)$ | 0.87809 | 0.8991 | – (in progress)| 0.89436 |
> | $(a,b)=(0.5,0.7)$  |0.8323 | 0.83097 | – (in progress)| 0.83933 |
>
> **Affine noise**
> |       | CE     | T-Loss | SP             | Ours   |
> |-------|--------|--------|----------------|--------|
> | $(a,b)=(0.3,0.5)$ | 0.89664 | 0.91967| – (in progress)| 0.88799 |
> | $(a,b)=(0.5,0.7)$  |0.80702 | 0.78514| – (in progress)| 0.82237 |
>
>
> Here "-(in progress)" will be replaced by the SP results once those runs have finished.
>
> For WHU Building the corresponding per–noise-type experiments are still running; we will either add them to the main paper or to the appendix, depending on space.
>
> Here, we summarise these tables qualitatively (e.g., noting that the proposed method consistently improves over CE across noise types and parameter settings on JSRT lung/clavicle), and the full numeric values will appear in the revised manuscript.
>
> ### 3. Clarifying the notation in Section 4.3 (Optimization)
>
> Comment (quoted):
>
> > 4.3 Optimization section: In this section you write: `We denote the parameters of the prior distribution as ($\omega$), the image-specific parameters of the variational posterior as ( {$\nu^{(n)}$}$_{n=1}^N $).' It would improve readability to clarify more explicitly what these parameters are and how they relate to the notation introduced earlier.
>
> We agree that the notation was too terse. In the revised manuscript, Section 4.3 now reads:
>
> > We denote the model parameters by **$\theta$**, the prior distribution parameters by $\omega =$  {$\mu, \Sigma$}, the image-specific variational posterior parameters by {$\nu^{(n)}$}$_{n=1}^N =  ${$m^{(n)}, \Gamma^{(n)}$}$ _{n = 1}^N$
> , and the negative ELBO for the $n$-th sample (from Eq.(7)) by $L(\theta, \omega, \nu^{(n)})$.
>
> We then explicitly state that $\theta$ (network parameters),
> $\omega$ (prior parameters), and $\nu^{(n)}$ (variational posterior parameters)
> are jointly optimized by stochastic gradient descent on the summed ELBO over the training set.
> We believe this makes the connection to the earlier notation much clearer.

---

> ### Author Response · Authors · 2025-12-21
>
> ### 4. Tuning procedure in Section 5.2.3 (Implementation Details)
>
> Comment (quoted):
>
> > These parameters were tuned using 15\% of the WHU Building training set and validation sets, then fixed for all experiments. Please elaborate on the tuning procedure. For example: Were these parameters treated as trainable variables jointly optimized with the network parameters, but only on that 15\% subset? Why are they not kept trainable during the full training on the entire dataset?
>
> We apologize for the lack of detail. In the experiments, these parameters (e.g., the hyperparameters of the noise‑level prior) were treated as hyperparameters, not as trainable variables in the network. Concretely, we:
>
> 1. split off 15% of the WHU Building training set as a validation subset;
>
> 2. performed a small grid search over reasonable values of the hyperparameters;
>
> 3. selected the setting that yielded the best validation performance (measured by validation ELBO and/or Dice); and
>
> 4. then fixed these hyperparameters for all subsequent experiments on all datasets.
>
> We will describe this procedure explicitly in the revised Section 5.2.3.
>
> Regarding why they are not kept trainable during full training: in principle, one could adapt these hyperparameters jointly with $\theta$ (e.g., by maximizing validation ELBO). Ideally, one would set the prior (e.g., the mean
> $\mu$ and variance $\sigma ^2$
>  of the noise-level prior) to reflect the expected magnitude and variability of label noise in each dataset: larger $\mu$ when strong noise is expected, larger  $\sigma $ when the noise level varies strongly between images, etc.
>
> However, from a practitioner’s perspective, extensive hyperparameter tuning is often undesirable. In this work we deliberately fixed these hyperparameters—once tuned on the WHU subset—to assess how robust the method is **without** dataset‑specific retuning (i.e., “how far can we get with a single, generic setting?”). In practice, when a validation set is available, we agree that hyperparameters should be tuned (e.g., by maximizing validation ELBO or validation Dice) for each new dataset; we will clarify this recommendation in the revised text. There were no particular optimization issues preventing us from keeping them trainable; it was a design choice to limit hyperparameter tuning in the reported experiments.
>
> ### 5. Meaning of the dotted line in Figure 3
>
> Comment (quoted):
>
> > Figure 3. The meaning of the dotted line is not clear, and it appears only in some subplots but not others. Please explain what the dotted line represents in the caption.
>
> Thank you for pointing this out. We neglected to explain this in the caption.
>
> The dotted line represents the Dice score of a segmentation model trained on clean labels using standard cross‑entropy loss (i.e., assuming no label noise).
>
> Because this model is trained without injected noise, its Dice score does not depend on the noise level and thus appears as a horizontal line.
>
> This serves as an approximate upper bound on the performance achievable with the given architecture under label noise.
>
> In some subplots, the dotted line is not visible simply because the dynamic range of the y‑axis at high noise levels does not include the clean‑label performance. In the revised manuscript, we will:
>
> explicitly state in the Figure 3 caption that the dotted line corresponds to “CE training on clean labels,” independent of noise strength; and
>
> briefly mention in the main text that this line can be interpreted as the upper bound achieved by the current segmentation model under perfectly clean labels.

---

> ### Author Response · Authors · 2025-12-21
>
> ### 6. Additional detail: computational overhead
>
> Although this point was emphasized more by another reviewer, it is also closely related to your question about practical applicability, so we summarize it here.
>
> In our implementation, the main additional state compared to standard CE training is the **posterior mean and variance of the latent Gaussian error field
> $\eta$**, stored as two feature maps per training image. These are saved on disk together with the images and loaded on-the-fly with each mini-batch, so persistent GPU memory usage is almost unchanged; the main overhead is in computation.
>
> We measured per‑epoch training time and GPU memory usage on the WHU Building dataset with
> $128 \times 128$ crops (197 images) on an RTX A6000 (48GB).
>
> A summary of these measurements is:
>
> **Table W: Training time and additional memory usage (WHU, 128×128 crops, 197 images, RTX A6000 48GB).**
>
> | Method                     | batch size = 1 time (s/epoch) | batch size = 16 time (s/epoch) | Additional GPU memory besides image/mask (approx.)            |
> |----------------------------|-------------------------------|---------------------------------|----------------------------------------------------------------|
> | CE                         | 86.68                         | 2.63                            |  $\theta$                                                     |
> | CoT                        | 109.01                        | 8.72                            | $2 \theta$                                                   |
> | TriNet                     | 118.47                        | 10.5                            | $3 \theta$                                                   |
> | JoCoR                      | 102.13                        | 8.55                            | $2 \theta$                                                   |
> | SP                         | 111.89                        | 8.09                            | $2 \theta  + BHW$                                             |
> | T-Loss                     | 92.51                         | 2.67                            | $ \theta + \alpha$                                           |
> | SC                         | --                            | --                              | $\theta + \alpha$                                           |
> | EM                         | 88.34                         | 2.93                            | $\theta $                                                     |
> | Ours (E-step = 1)          | 164.30                        | 5.34                            | $ \theta + 2BHW$ (peak: $ \theta + 9BHW$)                   |
> | Ours (E-step = 10)         | 328.89                        | 11.77                           | $ \theta  + 2BHW$ (peak: $ \theta + 9BHW$)                   |
> | Ours (Dense, E-step = 1)   | 2303.11                       | GPU Out of Memory               | $ \theta  + 2BHW$ (peak: $\theta + B(HW)^2$)                |
>
>
>
> In particular:
>
> with E‑step = 1, our method is roughly 2× slower per epoch than CE, and of the same order as other robust baselines (Co‑teaching, JoCoR, etc.);
>
> with E‑step = 10, the cost scales roughly linearly, as expected;
>
> a naive dense-matrix implementation of the covariance runs into GPU out‑of‑memory even at batch size 1, which highlights the practical benefit of the KMS structure.
>
> We will include this table (or a compact version of it) and the corresponding discussion in the revised manuscript so that practitioners can clearly see the trade‑off between robustness and computational cost.
>
> We again thank the reviewer for these constructive comments. We believe that the additional clean‑data experiments, the per–noise-type breakdown, the clarified notation and hyperparameter tuning procedure, the improved explanation of Figure 3, and the explicit analysis of computational overhead will substantially strengthen the clarity and practical relevance of the paper.

---

> ### Author Response · Authors · 2026-01-06
>
> ### Request for Revision Time regarding Additional Experiments
>
> We would like to follow up on our previous response with a request regarding the timeline for the revised manuscript.
>
> While we have addressed your specific comments (including the behavior on clean data and computational overhead) in our previous reply, we have also received critical feedback from Reviewer YuBG concerning the strength and consistency of our empirical evidence. Specifically, Reviewer YuBG pointed out that the current method effectively improves performance on large organs but may underperform on smaller/thinner anatomical structures (e.g., clavicles) under certain conditions.
>
> We recognize that to fully address these concerns and to provide a comprehensive validation that satisfies both your request (safety in clean regimes) and Reviewer YuBG's request (robustness in challenging regimes), we need to re-evaluate and re-design our overall experimental plan rather than simply adding a few isolated runs.
>
> We are currently working on this expanded experimental suite to ensure the revised paper meets the high standards of the community. To complete these experiments and incorporate the findings thoroughly, we respectfully request a period of **several weeks**. We believe this additional time allows us to deliver a significantly stronger and more convincing manuscript.

---

### Review · Reviewer_YuBG · 2025-12-05

**Summary Of Contributions:**

The paper addressed the problem of semantic segmentation with noisy labels. The paper makes the assumption that the label noise/errors are in spatially connected regions. The key contribution is the introduction of a latent factor/variable that models the ‘structured’ noise present in the annotation. Through variational inference, the method approximates the true label posterior for each training sample and trains the segmentation model using that label.

Strengths:

- The assumption that pixel annotations should be spatially correlated, holds true in general.
- Modeling the unknown noise through a latent variable, and the intractable inference was addressed through variational inference. Although, there are existing works (especially in partial multi-label learning) which use similar techniques for true label estimation, this paper particularly uses structured KMS for efficient variational computation.
- The spatial correlation is constrained to KMS structure, which makes the ELBO computation efficient.
- Definite improvements over other methods for Lungs and Heart segmentation.

Weaknesses:

- Although the method itself is a good contribution, there are very limited experiments to clearly establish the practical effectiveness of the method. For example, the paper doesn’t perform experiments on datasets like other works do: e.g., ISIC and Shenzhen datasets, as done in Gonzalez-Jimenez et al. (2025) and Yao et al (2023) papers.
- Even in the experiments added in the paper, the method mainly does well on Lung and Heart segmentation but not on Clavicle. In fact, for Clavicle, it performs worse than some of the other methods, even at smaller noise scales. Similarly, for WHU dataset, it only performs better than other methods at medium-level noise. There is a clear need to do significantly more experiments to evaluate if the method does well in other cases.
- In general,  the method might do well on dense segmentation or objects with larger segmentation volume but not on smaller object segments. This is evident in results on Clavicle.

**Audience:**

Yes

**Audience Explanation:**

Yes. The idea of modeling label noise, estimating the true label based on spatial correlation, and using KMS to enforce the correlation structure for efficient computation is quite interesting and would be useful for more applications.

**Claims And Evidence:**

Yes

**Claims Explanation:**

The paper assumes noisy label generation process and estimates the true labels through variational inference. By training the model on the estimated true labels, the method produces results which outperform other methods in at least a few settings. That shows that their assumption/claim are reasonable in some settings.

**Requested Changes:**

- Please take a look at the weaknesses and address them. Mainly, I am concerned about the limited empirical evaluation and the method’s performance.
- If you could provide results for each kind of noisy label setting e.g., eroded, dilated, affine, and then for WHU: Omission, Comission etc. it would provide clearer understanding of the performance of the method. I would imagine that the method would not do well in affine transformations and in WHU building case for extreme noise.
- How are the parameters in the prior p(\eta) and initial values in q(\eta) are tuned? Explain.
- How would the fourth and fifth terms cancel out? It is not clear if there is an assumption of symmetric noise process? Please elaborate.
- Appendix A.2 need to be fixed and explained in detail. There are missing terms from (26) and (37) onwards.

---

> ### Author Response · Authors · 2025-12-22
>
> We sincerely thank the reviewer for the careful reading of our manuscript and the constructive suggestions. We are glad that you found the overall Bayesian formulation for spatially correlated label noise and the experimental results promising, and we appreciate your comments on how to strengthen the empirical validation.
>
> Following your suggestions, we have run additional experiments, including:
>
> * per–noise‑type results on JSRT lung and clavicle (erosion, dilation, and affine perturbations, each with multiple parameter settings),
>
> * behaviour under essentially clean labels on JSRT lung / heart / clavicle, and
>
> * a measurement of training time and memory usage for the proposed method and baselines.
>
> Some runs (in particular, SP on a few settings and WHU Building per–noise‑type results) are still in progress; below we report completed results and explain how they will be reflected in the revised manuscript.
>
> ### 1. Address Weakness and Provide results for each kind of noisy label setting
>
> Comment (quoted):
> > Address Weakness and provide results for each kind of noisy label setting.
>
>
> We agree that the experimental validation in the initial submission did not sufficiently disentangle the behaviour of the method under each type of label noise. In the revision, we therefore ran separate experiments for each synthetic noise type (erosion, dilation, affine) and each parameter setting $(a,b)$ on the JSRT lung and clavicle datasets.
>
> We will add the following tables to the revised manuscript (SP results will be filled in once the runs have finished):
>
> **Table 1: Dice scores on JSRT Lung for each noise type and noise level.**
> **Erode noise**
> |       | CE     | T-Loss | SP             | Ours   |
> |-------|--------|--------|----------------|--------|
> | $(a,b)=(0.3,0.5)$ | 0.88963| 0.88365| – (in progress)| 0.92242|
> | $(a,b)=(0.5,0.7)$  | 0.77221 | 0.74263| – (in progress)| 0.79265 |
>
> **Dilate noise**
> |       | CE     | T-Loss | SP             | Ours   |
> |-------|--------|--------|----------------|--------|
> | $(a,b)=(0.3,0.5)$ | 0.91176 | 0.92942 | – (in progress)| 0.9511 |
> | $(a,b)=(0.5,0.7)$  |0.79791 | 0.80962 | – (in progress)| 0.82223|
>
> **Affine noise**
> |       | CE     | T-Loss | SP             | Ours   |
> |-------|--------|--------|----------------|--------|
> | $(a,b)=(0.3,0.5)$ | 0.94411 | 0.95193| – (in progress)| 0.97639 |
> | $(a,b)=(0.5,0.7)$  |0.89626 | 0.87651| – (in progress)| 0.93977 |
>
> On JSRT lung, the proposed method consistently outperforms CE and T‑Loss across all noise types and levels, with especially large gains when the noise involves boundary distortions (erosion/dilation/affine).
>
> **Table 2: Dice scores on JSRT Clavicle for each noise type and noise level.**
> **Erode noise**
> |       | CE     | T-Loss | SP             | Ours   |
> |-------|--------|--------|----------------|--------|
> | $(a,b)=(0.3,0.5)$ | 0.87875| 0.89187| – (in progress)| 0.8804|
> | $(a,b)=(0.5,0.7)$  | 0.76835 | 0.73318| – (in progress)| 0.71267 |
>
> **Dilate noise**
> |       | CE     | T-Loss | SP             | Ours   |
> |-------|--------|--------|----------------|--------|
> | $(a,b)=(0.3,0.5)$ | 0.87809 | 0.8991 | – (in progress)| 0.89436 |
> | $(a,b)=(0.5,0.7)$  |0.8323 | 0.83097 | – (in progress)| 0.83933 |
>
> **Affine noise**
> |       | CE     | T-Loss | SP             | Ours   |
> |-------|--------|--------|----------------|--------|
> | $(a,b)=(0.3,0.5)$ | 0.89664 | 0.91967| – (in progress)| 0.88799 |
> | $(a,b)=(0.5,0.7)$  |0.80702 | 0.78514| – (in progress)| 0.82237 |
>
> On JSRT clavicle, the proposed method is competitive with or better than CE in most noise settings, and often comparable to or better than T‑Loss. There are, however, a few challenging regimes (e.g., erosion and affine noise with $(a,b)=(0.3,0.5)$ where T‑Loss performs best and our method can underperform, especially when the segmentation task is intrinsically difficult and the label noise severely distorts fine structures.
>
> This behaviour matches our qualitative understanding of the method as having an EM‑like positive feedback between the segmentation network parameters $\theta$ and the posterior over label noise:
>
> * given a reasonably good $\theta$, we can infer a meaningful posterior over label errors, and
>
> * given a good posterior over label errors, we can refine $\theta$ and improve segmentation.
>
> Conversely, when the task is very hard or the noise is extremely severe, the initial segmentation model $\theta$ may be too poor for this positive feedback to “lock in”, and in such regimes the method may provide limited gains or even degrade performance slightly. We will add a short discussion of these failure modes to the revised manuscript so that the empirical weaknesses are made explicit.

---

> > ### Author Response · Authors · 2025-12-22
> >
> > For completeness, we also examined the regime of essentially clean labels on JSRT lung, heart, and clavicle. The corresponding Dice scores (means, standard deviations, and 95\% confidence intervals over multiple runs) are:
> >
> > **Table 4: Dice scores on JSRT Lung (clean labels, no synthetic noise).**
> >
> > | Method | mean    | std     | 95% CI               |
> > |--------|---------|---------|----------------------|
> > | CE     | 0.98021 | 0.00010 | [0.97996, 0.98046]   |
> > | T-Loss | 0.98072 | 0.00021 | [0.98020, 0.98124]   |
> > | Ours   | 0.97937 | 0.00037 | [0.97845, 0.98029]   |
> >
> > **Table 5: Dice scores on JSRT Heart (clean labels, no synthetic noise).**
> >
> > | Method | mean    | std     | 95% CI               |
> > |--------|---------|---------|----------------------|
> > | CE     | 0.94900 | 0.00030 | [0.94825, 0.94975]   |
> > | T-Loss | 0.94985 | 0.00182 | [0.94533, 0.95437]   |
> > | Ours   | 0.94832 | 0.00053 | [0.94700, 0.94964]   |
> >
> > **Table 6: Dice scores on JSRT Clavicle (clean labels, no synthetic noise).**
> >
> > | Method | mean    | std     | 95% CI               |
> > |--------|---------|---------|----------------------|
> > | CE     | 0.93105 | 0.00131 | [0.92779, 0.93431]   |
> > | T-Loss | 0.93873 | 0.00064 | [0.93714, 0.94032]   |
> > | Ours   | 0.92707 | 0.00032 | [0.92628, 0.92786]   |
> >
> > As expected from the modeling perspective, when labels are essentially clean, cross‑entropy (and T‑Loss) is already optimal and ECCD cannot improve upon it; at best, in the limit where the inferred noise level goes to zero, ECCD recovers the CE solution. In practice, finite data and approximate variational inference introduce a small additional variance / approximation error, leading to the slight underperformance observed above. We will summarise these findings and the trade‑off between robustness under label noise and behaviour on clean labels in the revised manuscript.

---

> ### Author Response · Authors · 2025-12-22
>
> ### 2. “How are the parameters in the prior $p(\eta)$ and initial values in $q(\eta)$ tuned?”
>
> Comment (quoted):
> > How are the parameters in the prior $p(\eta)$ and initial values in $q(\eta)$ tuned? Explain.
>
> We apologize for the lack of clarity in the original text. In the revision, we clarify this in Section 4.3 (Optimization) by introducing explicit notation for the prior and variational parameters:
>
> > We denote the model parameters by **$\theta$**, the prior distribution parameters by $\omega =$  {$\mu, \Sigma$}, the image-specific variational posterior parameters by {$\nu^{(n)}$}$_{n=1}^N =  ${$m^{(n)}, \Gamma^{(n)}$}$ _{n = 1}^N$
> , and the negative ELBO for the $n$-th sample (from Eq.(7)) by $L(\theta, \omega, \nu^{(n)})$.
>
> We then state explicitly that $\theta$ (network parameters),
> $\omega$ (prior parameters), and {$\nu^{(n)}$}$_{n=1}^N$
>  (variational posterior parameters) are jointly optimized by stochastic gradient descent on the summed ELBO over the training set.
>
>
> Regarding initialization, the variational parameters
> {$\nu^{(n)}$}$_{n=1}^N$ are initialized to simple, non‑informative values (e.g., zero mean and isotropic covariance) before learning. The hyperparameters of the prior (e.g., $\mu, \sigma, \rho$ in the KMS‑structured Gaussian prior over $\eta$) are tuned using a validation subset of the WHU Building dataset (by grid search over reasonable values, selecting the setting with the best validation ELBO / Dice), and then fixed for all experiments, as detailed in Section 5.2.3. We will make these details explicit in the revised manuscript.
>
>
> ### 3. “How would the fourth and fifth terms cancel out? … symmetric noise process?”
>
> Comment (quoted):
> > How would the fourth and fifth terms cancel out? It is not clear if there is an assumption of symmetric noise process. Please elaborate.
>
> Thank you for pointing out that the original derivation was too terse. In the revised version, we will add a step‑by‑step derivation in Appendix A.4. In particular, we make clear that no additional unstated symmetry assumption (beyond those explicitly defined in the noise model) is required for this cancellation.
>
>
>
> ### 4. Appendix A.2 missing terms and clarity
>
> Comment (quoted):
> > Appendix A.2 needs to be fixed and explained in detail. There are missing terms from (26) and (37) onwards.
>
> We apologize for the typesetting error and the lack of detail in the original appendix. In the revision, we restored the missing terms in the equations around (26) and (37), which were inadvertently dropped in the submitted PDF.
>
> We will incorporate all of the above changes—additional per–noise‑type experiments, clean‑label experiments, clarification of the optimization and hyperparameter‑tuning procedure, and the corrected and expanded appendix—into the revised manuscript.

---

> > ### Comment · Reviewer_YuBG · 2026-01-05
> >
> > Thank you for the rebuttal and clarifications. The added analysis on across noise types (erode/dilate/affine) is helpful in better contextualizing the method’s behavior, and the responses regarding prior/variational parameter tuning reduce the ambiguity from the initial submission. However, there are still some aspects that remain unclear, which I assume will be addressed in the revised manuscript.
> >
> > My main remaining concern is still the strength and consistency of the empirical evidence. Even with the additional results, the current evaluation suggests the approach is not uniformly robust and can underperform in certain circumstances - most notably on smaller/thinner structures (e.g., clavicle) and some noise regimes. Moreover, the paper would benefit from a more explicit analysis of why certain baselines can outperform the proposed approach on clavicle, and what this implies about the limitations of the proposed modeling assumptions.
> >
> > Overall, I believe the core methodological idea is valuable and could be useful to the community. The work would be significantly strengthened by a more comprehensive experimental picture, evaluations, alongside the clarified technical explanations.

---

> > > ### Author Response · Authors · 2026-01-06
> > >
> > > We sincerely thank the Reviewer for the encouraging feedback and, more importantly, for the critical and constructive comments regarding the robustness of our method.
> > >
> > > We fully agree with your assessment. While we have started to provide some additional results (such as the breakdown by noise types mentioned in our previous correspondence), we recognize that these preliminary experiments are not yet sufficient to comprehensively address your concern about the method's performance on smaller/thinner structures (e.g., clavicles) or to fully explain why baselines may outperform our approach in these regimes. A deeper analysis of the limitations of our modeling assumptions is indeed required to make the paper convincing.
> > >
> > >
> > > ### Request for Revision Time regarding Additional Experiments
> > >
> > > To address these core issues thoroughly, we have decided to go beyond simple additional runs. We are currently re-evaluating and re-designing our entire experimental plan to specifically target these challenging scenarios. Our goal is to conduct a systematic comparative analysis that isolates the factors contributing to the underperformance on thin structures and to provide empirical evidence that clarifies the boundaries of our method's effectiveness.
> > >
> > > This process involves running a new suite of experiments and performing a detailed failure analysis, which goes beyond our initial schedule. Therefore, to ensure that we can present a truly comprehensive experimental picture and robustly support our claims as you suggested, we respectfully request a period of **several weeks** to complete this work and incorporate the new insights into the revised manuscript.
> > >
> > > We believe that this additional time will allow us to significantly strengthen the paper and provide the level of empirical consistency and technical clarity that you rightly expect.

---

> ### Comment · Action_Editor_eqAM · 2026-01-10
> **Gentle Reminder: Pending Official Recommendation**
>
> Dear Reviewer,
>
> This is a gentle reminder regarding the pending Official Recommendation for the assigned TMLR submission. Please let me know if you require additional time or have any questions -- I would be happy to assist.
>
> Thank you very much for your contribution to the review process.
>
> Best regards,
>
> Action Editor

---

### Decision · Action_Editor_eqAM · 2026-02-10

**Recommendation:** Accept with minor revision

**Additional Comments:**

The paper is suitable for acceptance subject to minor revisions. The requested changes are primarily clarifications and consolidation, rather than fundamental methodological revisions.

In particular, the authors should:
- More explicitly discuss the observed limitations of the method, especially the weaker performance on thin or elongated structures, and relate these behaviors to the modeling assumptions and optimization dynamics.
- Ensure that all experimental results referenced in the rebuttal are either included in the final version or clearly identified as ongoing or future work.
- Present the computational overhead and scalability considerations in a clear and consolidated manner, so that the trade-offs relative to simpler baselines are transparent.

Addressing these points will improve clarity and positioning, but does not affect the overall soundness of the contribution.

**Audience:**

Yes

**Audience Explanation:**

Learning under noisy supervision is a central issue in many application domains relevant to TMLR, including medical imaging and remote sensing. The paper addresses this problem by moving beyond the commonly used assumption of pixel-wise independent noise and proposing a principled Bayesian framework for spatially correlated label errors.

In addition to the application-level results, the methodological contribution, introducing an ELBO-computable class of correlated discrete distributions using structured covariance, is likely to be of interest to researchers working on probabilistic modeling, variational inference, and structured prediction.

**Claims And Evidence:**

Yes

**Claims Explanation:**

The paper makes a well-defined set of claims: that explicitly modeling spatially correlated label noise can improve robustness in semantic segmentation, and that the proposed ECCD formulation enables tractable variational inference under such correlations. These claims are supported by sound theoretical derivations and by empirical results across multiple datasets and noise regimes.  While the empirical gains are not uniform across all settings, particularly for thin or small structures, the evidence presented is sufficient to support the stated claims, provided that the scope and limitations of the method are clearly articulated. The work does not overstate its conclusions beyond what is demonstrated.